# Recent Progress in Conducting Polymer Composite/Nanofiber-Based Strain and Pressure Sensors

**DOI:** 10.3390/polym13244281

**Published:** 2021-12-07

**Authors:** Loganathan Veeramuthu, Manikandan Venkatesan, Jean-Sebastien Benas, Chia-Jung Cho, Chia-Chin Lee, Fu-Kong Lieu, Ja-Hon Lin, Rong-Ho Lee, Chi-Ching Kuo

**Affiliations:** 1Institute of Organic and Polymeric Materials, Research and Development Center of Smart Textile Technology, National Taipei University of Technology, Taipei 10608, Taiwan; anloga947715@gmail.com (L.V.); manikandanchemist1093@gmail.com (M.V.); benas.jeansebastien@gmail.com (J.-S.B.); 2Department of Physical Medicine and Rehabilitation, Cheng Hsin General Hospital, Taipei 11220, Taiwan; ch1931@chgh.org.tw; 3Department of Physical Medicine and Rehabilitation, National Defense Medical Center, Taipei 11490, Taiwan; 4Institute of Electro-Optical Engineering, National Taipei University of Technology, Taipei 10608, Taiwan; jhlin@ntut.edu.tw; 5Department of Chemical Engineering, National Chung Hsing University, Taichung 40227, Taiwan; rhl@dragon.nchu.edu.tw

**Keywords:** conducting polymers, polymer composites, nanofibers, strain sensors, pressure sensors

## Abstract

The Conducting of polymers belongs to the class of polymers exhibiting excellence in electrical performances because of their intrinsic delocalized π- electrons and their tunability ranges from semi-conductive to metallic conductive regime. Conducting polymers and their composites serve greater functionality in the application of strain and pressure sensors, especially in yielding a better figure of merits, such as improved sensitivity, sensing range, durability, and mechanical robustness. The electrospinning process allows the formation of micro to nano-dimensional fibers with solution-processing attributes and offers an exciting aspect ratio by forming ultra-long fibrous structures. This review comprehensively covers the fundamentals of conducting polymers, sensor fabrication, working modes, and recent trends in achieving the sensitivity, wide-sensing range, reduced hysteresis, and durability of thin film, porous, and nanofibrous sensors. Furthermore, nanofiber and textile-based sensory device importance and its growth towards futuristic wearable electronics in a technological era was systematically reviewed to overcome the existing challenges.

## 1. Introduction

Recent trends have evolved to numerous stimuli-based electronic appliances, especially in terms of understanding human biological and physiological fitness. Numerous credible establishments have achieved this via a spectrum of factors that include materials, design, fabrication, processing, and stability conditions. An emerging demonstration of efficient high resolution sensors in recent years has caused significant progress in elucidating the chemical, biochemical, biological, toxic, organic effluents, inorganic metals, strain, and pressure signals through chemical bonding interactions, aggregation-induced, optical, colorimetric, and structural color responses [1,2,3,4]. Among which, strain and pressure sensors are highly desirable for assessing human health status, object detection, heartbeat, muscle, body motion, respiratory detection, and infant–elderly comfort monitors [5,6,7,8]. Although the evolution of strain and pressure sensors has been rapid, the increasing progression towards the achievement of 5S, i.e., sensitivity, selectivity, size ability, stability, and scalability is evident. The upcoming surge in elevating the eco-friendliness and bio-compatibility remains as another crucial task for the researchers to benefit the human society. Material choice ranging from rigid, flexible, and stretchable polymers, along with conductive nanostructures, including nanoparticles, nanowires, nanorods, nanoflakes, nanosheets, and nanofibers, pave the way to upgrades in the sensors’ real-time suitability [9,10,11].

Polymer composites not only limit their arms to sensors and energy generators, but they also govern breakthroughs in fields such as light-emitting diodes (LEDs), lasers, solar cells, field effect transistors, memory devices, and soft robots [12,13,14]. For instance, polymeric composite light emissive layers empower the efficiency and stability of the LEDs [15,16]. Recently, polymeric interlayers proved to be effective in brightening LED device performance. Kuo et al. designed the stretchable perovskite LEDs composed of tri-composite perovskite polymeric emissive layers that can retain its luminance even under strained conditions [17]. Following this, polymeric interface-assisted grain control process sorts out the efficiency and the air and humidity’s environmental stability issue synchronously to obtain the bright, emissive layers [18]. The performance of solar cells has flourished significantly, achieving better stability and power conversion efficiency, as it curtails defect states and promotes grain growth [19,20]. Recent reviews pose a better insight on how such polymeric composites have brought enlightenment in the field of light emitters and solar cells [21,22]. Persistent efforts and attempts dedicated to developing highly efficient optoelectronic devices and several breakthroughs have been convincingly made to leverage the polymer composites [23,24].

Apart from optoelectronics, polymer composites and their influence in generating gamut of strain and pressure sensors is appreciable. Energy generators play a critical role in enhancing the sensors’ utilities, as they reduce the power consumption, i.e., self-powered complexities, in coupling the energy source or other energy storage components [25]. Both in terms of sensing and energy generators, it is inexorable to neglect the role of nanofibers and the functionality inclusion into smart textiles, experiencing explosive outgrowth in wearable electronics [26,27,28]. This review highlights the importance of polymer composites and nanofibers in the fabrication of strain and pressure sensors. We herein review the challenging aspects of designing highly responsive sensors through facile synthetic strategies, sensor fabrication, mechanistic aspects, various sensing modes, and their operational ranges. Recently, reviews on conjugated copolymers and nanofibers in the field of sensing various factors including metal ions, pH, temperature, and humidity imparts knowledge on colorimetric and fluorometric optical visible responses [29,30]. Conducting polymers playing a credulous role in the design of biosensors and recent reviews highlights their significance in the preparation and properties, of bioapplications [31,32]. This review details the classification of strain and pressure sensor fabrication and its importance in overcoming existing sensory challenges such as sensitivity, operating range, durability, response time, stability, and their adaptability to sensing environments. The basic mechanism for operating the strain sensors falls within the fabrication of conductive networks and their regulated conductive responses with respect to applied strains [33,34]. Several conducting polymer composites, conductive hybrid networks, functionalized composites, co-polymeric systems, 2D nanostructures, structural modifications, and nanofibrous architectural strategies are moving progressively towards the attainment of crucial characteristics in strain and pressure sensors [35,36,37,38,39].

## 2. Sensor Device Performance Characteristics

Sensors are generally denoted as devices that can brilliantly respond to analyte or external stimuli selectively (pollutants, pH, temperature, humidity, etc.) with good sensitivity and linear response. In particular, strain sensors respond intelligently in response to various physical stimuli, such as tensile, compression, twist, etc. The strain sensor performances can be evaluated and compared effectively with their sensitivity, operational range, response and recovery time, stability, and reusability [40,41].

*Sensitivity*: A variation of electrical sensory signals in accordance to the strain experienced can be termed as sensitivity. The majority of the strain sensor response can be widely recorded as electrical outputs and, in a few studies, it has been analyzed with optical measurements. The gauge factor (GF) value can be derived with the help of the following equation [42,43],
$$\mathrm{GF}=\Delta R/R_{0}$$
where R is the resistance at the applied respective strain or pressure, R_0_ is the initial resistance of the sensor device. ΔR is the difference between initial and final resistance under strain or pressure condition.

*Operational range*: A liner plot of strain sensor response in a wide working range allows the user to correlate, and it is easier to fix the standards for the error-free strain monitoring.

*Response time*: The time at which the sensor records its electrical (resistance R) or optical response on reaction with the strain imposed (expressed as Δt_1_ in Figure 1a). Generally, lower response time sensors are highly valued, as they offer instant outputs without any processing delays.

*Recovery time*: The time at which the sensors revert to their original (resistance R_0_) state after relaxation (expressed as Δt_2_ in Figure 1a). The recovery time is also considered while employing the strain sensors, as it can delay the output data collection. A sensor should possess a lower response and recovery time when assessing human heart beats and human physiological signals, pronouncing the dynamic characteristics of the wearable e-skins.

*Stability*: The stability of the strain sensors threatens the deployment of the wearable devices in real-time, and it delays the evolution of the internet of things. Rapid human muscle movements and blood flow necessitates the demand for the stable recovery of strain-sensing devices. Hysteresis (H) is the sensory response difference between the initial and successive strain cycles (Figure 1b). For the stable strain sensor operation, the electrical hysteresis value should be minimum, and hysteresis differences should be insignificant. Electrical response lifetime and storage stability are other credulous factors that have to be taken into account in order to reach industrialization and commercialization.

*Reusability*: Reusable strain sensor with negligible hysteresis elevates the quality of the prepared strain sensors. Moreover, long-term operational stability studies have reached audience requirements in terms of connectivity and self-powered devices.

## 3. Strain Sensors’ Working Mode

The strain sensor’s working mechanism follows the resistive type, and there exists two probabilities: either a decrease or increase of electrical resistance depending on the degradation and establishment of conductive percolation networks upon tensile or compressive deformations. More concisely, tensile strain on the strain sensors cause strain gradients. Under smaller strains, the conductive fillers remain within the percolative network, resulting in negligible electrical signal variations. Whereas, upon applying successive improved strains, tunneling paths between the conductive networks gradually fail, and their electrical influences are predominantly noticeable in the form of electrical resistance increments. On gradual relaxation, the strain sensors re-execute their threshold in reforming the conductive networks, eventually contributing to their initial electrical resistance form [44,45]. The prime role of recovery in a strain sensor is largely dependent on the elastic mechanical behavior of the chosen elastomeric substrate [46]. In some cases, the hysteresis or recovery ratio of the sensors cannot fall within the desired range of strain sensing, which highlights the vital contribution of energy-dissipating molecular chains in the strain sensor fabrication [47,48]. Therefore, recovery ratio and relaxation factors of the sensory frameworks serve as crucial factors, and give rise to hysteresis in the fabricated strain sensor. In recent years, unique conductive structures and geometrical features such as morphology and pattern design have been deployed to effectively reduce the hysteresis of the strain sensors. Figure 2a–e resembles the simultaneous conductivity path breakage with respective strain increments. Figure 2f portrays the resistance–strain response curve, which reveals the resistance increments according to the straining extent.

The preliminary studies carried with stretchable matrix and the surface nanostructure formation follows the general concept for the strain sensor fabrication. A step forward made by Choi et al. to chemically modify the elastomeric substrate and such modifications promisingly works on widening the sensing range and its sensing capability. a conductive hybrid, formed with better dispersion among silver nanowires (AgNWs) and carbon black, aids in monitoring the human gestures [49]. The combination of interior and exterior conductive fillers through a multifilament approach significantly governs the sensing spectrum [50]. Another approach works similarly via intensive interpenetrating conductive nanostructures, which find a good fit for the finger-bend motion detections [51]. Sun et al. designed the wrinkle-assisted crack structures, taking advantage of Young’s modulus and elasticity mismatch between the elastomer and conductive layers [52]. In addition, lowering the electrical response hysteresis was done by patterning and structural modifications in the sensing platform [53,54]. More insights can be found in the following sections through a number of strategies.

Another mode of strain sensing is the compressive type, in which the sensors’ frameworks are subjected to pressure that cause subsequent strain onto the sensor platform (Figure 2g). The compressive strain sensor working mechanism and sensory response remain in contrast to the typical tensile strain sensor. In other words, compressive strain sensors work apparently as pressure sensors. Compression process facilitates the conductive filler networks to contact each other, surpassing the airy voids. The active airy contact points connect the conductive nanostructures, eventually forming the basis for electrical resistive changes. As the conductive point increments promisingly shift due to prominent strain effects, they form the basis for a resistive response plot. The resistance shifts downwards due to the improved conductive percolative networks causing electrical outputs (Figure 2h). Attempts to create porous structures using biologically derived substrates, commercial rough structures, melting and solubilizing of the sacrificial materials, etc., resulted in the successful figure of merits for the pressure sensors [55,56,57].

Many porous structures were configured rationally to find the potential in achieving better working range and durability. In addition, hydrogel- and nanofibrous-structured sensors have been growing rapidly due to their facile processing, scalability, and stable integration.

## 4. Pressure Sensors’ Working Mode

Pressure sensors respond intelligently with the electrical signals in response to the applied pressure. Pressure sensors have been developed into a multidisciplinary field of research, as they interconnect material synthesis, processing, fabrication, system engineering, and signal acquisition, including new technological upgrades such as artificial intelligence and the internet of thing [58,59,60]. Recent trends and a good deal of research interests have evolved with the active involvement of metallic nanostructures, metallic hybrids, carbonaceous material-based sensing, etc., [61,62,63]. Conducting polymer and its composite-based pressure sensors, including healthcare, electronic skin, artificial prosthesis, and human–machine interface, are presented, along with successful achievements in terms of improving the sensitivity, sensing range, and hysteresis reduction. Lastly, we review the future directions required to achieve scalability, facile fabrication, and greener processes for pressure sensor fabrication. The fundamentals of understanding the pressure sensing working mechanism can be broadly classified into capacitive, resistive, triboelectric, and piezoelectric sensors.

Capacitive pressure sensors convert mechanical pressure signals to electrical capacitive signals. The working mode is presented in Figure 3a. The capacitance is expressed by the equation *C* = *ε*_0_*ε*_r_*A*/*d*, where *ε*_0_ is the vacuum permittivity (8.854 × 10^−12^ F/m), *ε*_r_ is the relative permittivity, *A* is the active overlap area, and *d* is the distance between the adjacent electrodes [64]. The capacitive changes are primarily influenced by two contributing factors, such as *A* and d. The active area depends on the compressive strain and forces applied onto the sensor, whereas *d* changes abruptly with respect to pressure extent. Capacitive type emerges from low modulus elastomer-based dielectrics, as it can deform readily, even with subtly applied pressures. This type of sensor progresses better in terms of sensitivity and response speed by engineering the micro-structured electrodes and dielectric architectures [65]. Resistive pressure sensors are another common type due to their simplified architecture and easy-to-track validating process. A change of resistance depicts the signals attained for the mechanical deformation’s degree. Encouraging the conductive networks by applying the pressure builds the resistive response [66]. Geometrical and electrical parameters collectively contribute to the resistive changes, i.e., in terms of geometric area (length and area) and electrical conductivity (Figure 3b). Resistive signal response enhancements are achieved with architectural modifications, and hysteresis reductions are accomplished with composite/hybrid and encapsulation techniques. The resistive signal working range is extended through metallic hybrids such as nanoparticles, nanowires, nanorods, and nanotubes. More in depth progresses are sequentially detailed in the upcoming subsections with the aim of achieving a good figure of merits. Recently, Ramanavicius et al. designed piezoresistive tactile sensors based on velostat composite polymer, and studied their mechanical loading behavior, such as their stress–strain characteristics [67].

Triboelectric pressure sensors are another class wherein the charge accumulation and transfer during various contact and frictional movements involve two dissimilar materials occurring and facilitating the electricity generation, causing stable electrical responses (Figure 3c). The triboelectric series possess commonly used materials, which functions as a positive and negative depending on the charge density factors [68,69]. The extent of positive and negative separation between the two selective materials greatly influences the triboelectric response generation. The relatively dynamic net charge accumulation and transfer process is also governed by several additional factors such as geometrical patterns, roughness, surface potential values, etc. The mode of operation varies with the number of electrodes, contacts, and its sensory performance, which are influenced by other dimensionality factors, and are addressed by various microstructural and energy-dissipative material integration [70].

Piezoelectric pressure sensors work rapidly in converting the dynamic pressures into electrical responses, and the most beneficial type is the self-powered type, such as triboelectric sensors. Figure 3d represents the positive and negative charge separation with the applied force, causing the reconfigured dipoles. Such reconfigured dipole intensity is quantified with charge values, which can be given, in short, as follows: *q* = *d*_33_*F*, where *q* is the separated charges, *d*_33_ is the piezoelectric strain constant, and *F* is the applied force. The effectiveness can be gauged with the presented q values [71,72]. Following the above strain and pressure sensing mechanism, conducting polymers and their composites are very promising in the fabrication and upgrading of the figure of merits. This review encompasses the recent trends of conducting polymers’ composite and its credulous role in elevating strain and pressure sensor performance.

## 5. Conducting Polymer and Its Composites

The discovery of conductive polymers started with the groundbreaking discovery that halogen doped polyacetylene (–CH=CH–)_n_ shows high electrical conductivity, which led to the 2000 Nobel Prize in Chemistry award. Conductive polymers gain great interest and, in particular, polyacetylene exhibits a good order of conductivity enhancements after treatment with iodine due to configurational changes [73]. Such achievements triggered the research towards the exploration of the fundamental understanding and applicability of conductive polymers. A major difference in conjugated polymers from saturated polymers is their bonding (sp^2^P_z_ hybridized carbon) giving rise to three σ-bonds and π-bonds. Conjugation that prevails in the conducting polymer is the major contributor in bringing the intrinsic conductivity to the polymeric matrix. Conjugation signifies the alternate single and double bond system, and such alternate bonds are vital in promoting the charge carrier mobility over the polymeric skeleton. Conductivity enrichment configured through strategies ranging from doping to morphological fine-tuning.

Polyacetylene, one of the conducting polymers, exhibits solvent-driven actuation ascribed to the volumetric polymeric chain changes in response to the solvent’s intake and exclusion. Depending on the substituent, alterations exists with considerable free volume gradients, which makes them function as sensors and actuators. Solvents’ intake, or, in other words, solvents’ swelling phenomena, changes the free volume molecular distances, which results in the colorimetric or fluorescence response [74]. Kuo et al. designed a series of colorimetric sensors through the utilization of chemosensory functionalities, which interact with the analytes, causing visible color transformations [75,76,77]. Controlling the conjugated polymeric alignment and geometry is highly desired, and is not particularly limited to sensory applications. Such controls and confinements are crucial in depicting the higher carrier mobility in transistors [78], fluorescence bioimaging [79], photocatalyst [80], solar cells [81], etc. Polyaniline (PANI) composites work effectively in proliferating the electron transfer, facilitating the ionic conductivity and catalytic activity synchronously, leading to a betterment in photovoltaic performance [82]. As is typically representative for conducing polymers, polypyrrole, PANI, polythiophene, and poly(3,4-ethylenedioxythiophene) (*PEDOT*) polymers were widely studied in sensory applications.

In general, conducting polymers experience a mutual balance between pros and cons regarding their utilization as strain and pressure sensors. The major advantage is their cost, facile synthetic process, non-toxic nature, tunable conductivity, etc., whereas the drawbacks associated with conducting polymers are their solvent incompatibility, processing, morphological agglomerates, non-uniform dispersion, and higher conductivity, etc.

## 6. Polypyrrole Sensors

Polypyrrole is use in targeted conductive-oriented applications due to its excellence in electrical conductivity, thermal stability, facile preparation, and environmental stability. Such intriguing characters of polypyrrole significantly aids in reaching the prerequisites for strain and pressure sensor fabrication.

Smart garment fabrication is often involved with conductive polymer and passivating protective layers. The involvement of protective shell layers substantially improves the usability. Similarly, the substrate materials serve as a platform for accommodating the stress, and it is crucial to satisfy the fatigue strain recovery rate. Loss of elongation and recovery during the repetitive strain process can potentially degrade the repeatability of the fabricated strain sensors. The ferric chloride oxidant-assisted vapor phase polymerization (VPP) technique offers better adhesion between polypyrrole and the natural rubber elastomer through a plasma activation process [83]. Yusuf et al. systematically studied the influence of different oxidants in lowering the electrical resistivity and elevating the thermal properties, suggesting that the prime role of ferric chloride in was achieving highly conductive polypyrrole [84]. Generally, VPP process is cheap and effective in delivering the smooth, uniform conductive polymers. Conductive polymer formation by VPP is governed by several factors such as monomer concentration, reaction time, temperature, and humidity of the reaction chamber. Most conventionally, oxidant impregnated elastomeric substrates are allowed to polymerize by passing the pyrrole monomer with optimized vapor pressure. In contrast, Park et al. proposed a wet on wet VPP method, (Figure 4a) wherein the polypyrrole coating is performed with wet substrate to avoid the bulk polymerization, resulting in impurity formation [85].

Porous polyurethane (PU) films developed using polyethylene glycol sacrificial polymer blending followed by dimethyl formamide (DMF) solvent etching. Porous PU facilitates the formation of interpenetrating conductive polypyrrole polymeric links, which can respond electrically while wearing it as a waistband to monitor human breathing [86]. Taking advantage of phase separation, commercial polyester textile polypyrrole polymerization carried out using a solution process which is facile and operable at room temperatures with lower sheet resistance (Figure 4b) [87]. Intriguingly, Polypyrrole composite cyclic loading and unloading energy dissipation factors remained almost neutral over consecutive 200 compressive strain cycles in comparison to the conventional strain sensor structures. Such stable energy dissipation unveils the potential of the as-fabricated sensor to long-term operability [88].

Although the sensitivity and durability results are encouraging and interesting, the sensing range achievement is still yet to be explored. Wan et al. fabricated a wide range of sensors operable between 0.1 and 200 N using the porous and uniform wrinkled structure of polypyrrole possessing that is effective when binding with a sugar-templated porous PU substrate. Unique micro-wrinkled structures, formed due to the incomplete swelling of the crosslinked elastomer during the controlled polymerization process. Interestingly, the micro-wrinkled sponge consumes lesser power consumption due to the reduced resistance afforded by the effective binding and uniformity of polypyrrole conductive networks [89]. Strain sensing sensitivity increments were obvious with pore size alterations, and has been evidenced in recent literature studies. Pore size can be tailored with a number of strategies, including concentration, freezing temperatures [90], polymeric templates [91], sacrificial particles [92], freeze drying [93], etc.

Traditionally, cotton fibers have been turned into strain sensors by coating, depositing ex-situ and in-situ polymerization of conducting polymers. Zhao et al. proposed a wash-durable respiratory monitor composed of tri-layered architecture (rib knitted cotton fabric/polypyrrole/PU), where PU coating played a critical role in promoting the repeatability and in resisting the hysteresis and washing perturbations [94]. However, the flexible fibers failed to fit the strain sensors, as the knitted loops, after exceeding the strain limit, did not revert to their original state, or lead to mechanical failure. Commercial polyester spandex-blended fibers can impart elastic deformational recovery to a good extent. Elastic characters accompanied by directional properties promotes such elastic commercial fibers for strain sensor fabrication. Following this, knitted nylon fabrics incorporated with iron oxidant via in-situ thermal hydrolysis treatment provides surface support to improve the polypyrrole loading. This effective surface functionalization is effective in improving the strain sensor working range to about 100% strain [95]. The sensing range was further widened, utilizing the tri-conductive composite that includes copper nanowires (CuNWs), polypyrrole, and quasi-spherical silver nanoflowers. The role of conducting polymer polypyrrole is to provide ligand to metal charge transfer contribution and to offer oxidation resistance character to CuNWs. Additional silver nanostructures support the mirco-crack propagation, leading to prolong, even at higher strain conditions (185% strain) with better sensitivity [96].

Pressure sensors repeatability is highly challenging because of the rigid polymeric backbones in the conducting polymer matrix and the mismatch between the elastic moduli. Excessive loading of conductive polymer reduces the flexibility and makes the entire sensor component fragile when straining. Avoiding such a dense conducting polymeric network can be of greater significance in attaining the tunable elastic moduli and such tunability benefit positively in sustaining stress and strains. Interconnected hollow spheres of polypyrrole enables the conducting polymer to deform and recover elastically with respect to external pressure relax cycles. Moreover, the device crosslinking secures the sensory performance, even over several months [97]. Another imperative factor in sensing aspects is identifying and neglecting the interference signals apart from pressure. Major influencing factors such as temperature and humidity seriously alter the sensory device’s performance, and accounts for the misleading erroneous results. Higher elastic modulus caused the conducting composite to decompose and delay its cyclability in sensory applications. Commercial Polydimethylsiloxane (PDMS) is well known for its mechanical flexibility and stretchability. Park et al. utilized the diluted PDMS to drop-cast over the conductive composite foam to retain the porous structure and impart better elastic nature to achieve good recoverability [98]. Following this, the achievement of hysteresis reduction with a uniform pore size and distributed foams was attained, and the study clearly reveals the distinct reduction in hysteresis percentage in sensory device performance. Additional chemical interactions confine the polymer chain sliding and displacements, thereby critically contributing to the effective reduction in hysteresis factors and coefficient of variation [99].

Apart from hysteresis and variance reduction, stability of polypyrrole is also considered for developing corrosion-free conductive films similar to conventional metal electrodes. Polypyrrole in chemical and electrochemical processes undergo considerable degradation, and it is contributed to by various factors including dopants, pH, dissolved oxygen, and temperature. More particularly, small dopants size selection is crucial for anion exchange because of the intrinsic higher mobility exhibited by smaller ions. In contrast, bulky steric dopants can support the cation exchange because of larger bulkiness controlled mobility, whereas the medium-sized dopants exhibit both anion and cation exchange behavior. Based on these versatile ion exchange characters, polypyrrole can be applied as an ion transport membrane for water purification [100]. Nonetheless, such ion exchange characters of doping anions with OH^−^ and the penetration of OH^−^ into the polypyrrole films is detrimental in supporting the degradation of polypyrrole in NaOH. Dopants of different anion types and their influence with distinct effects on the chemical composition and structure of polypyrrole films during the degradation process in alkaline medium were utilized to study the degradation kinetics of various doped polypyrrole films [101].

Very recently, Lee et al. utilized a plant-derived, bio-renewable curcumin template to overcome the cycling instability in supercapacitors. This cyclic instability arises due to the gradual swelling and shrinking of the polypyrrole backbone on the charging/discharging process. Nano-micro structuring can lead to effective charge transport and better electrolyte exposure. A scalable eco-friendly method to achieve the cyclic stability by accommodating the morphological swelling shrinking process remains appealing [102].

Polypyrrole transformed gradually into biological applications due to their non-toxic, unremitting stability, biocompatibility, etc. Very recently, biodegradable eco-friendly hanji papers crosslinked with phytic acid and the resulting surface demonstrated that polypyrrole exhibits better electrical stability, thermal properties, and near infra-red photothermal stability [103]. Conductive polymers are useful for guiding cell adhesion and proliferation. However, polypyrrole exhibits a brittle nature, and scaffolding is usually done with flexible elastomers. Following this, Shi et al. explored the potential of polypyrrole in the proliferation of PC12 cells with nanoporous cellulose gels/polypyrrole nanoparticle composite. The as-prepared composites were tuned into aerogels by supercritical carbon dioxide to form mechanically durable porous structures with enhanced conductivity, showing promise in nerve tissue regeneration [104]. Cell growth control is another crucial sector and, in recent years, such growth acceleration has been made feasible with electrical stimulation. Lee et al. prepared a conductive dual network hydrogel made of polypyrrole, significantly increasing its role in promoting the mechanically robust, tough networks [105]. Polypyrrole coating effectively restricts the dissolution of transition metal oxides in electrodes, thereby improving the cycling stability of supercapacitors [106]. Overall, Polypyrrole conductivity, processing factors, cyclability, and stability towards an alkaline medium can be explored further to gain a better insight into fine-tuning sensing attributes with the aim of creating cytocompatible mechanically robust wearable sensors.

## 7. Polyaniline Sensors

Polyaniline (PANI) has received attention amongst other conductive polymers due to the facile synthesis, cost factors, tunable conductivity, electrical stability, environmental stability, and their unique doping de-doping mechanism [107]. PANI polymer in pristine form possess only benzenoid rings. Oxidized forms are identified with a number of quinoid rings, and, depending on the number of quinoid rings, it is termed protoemeraldine, emeraldine, nigraniline, or pernigraniline. Protoemeraldine has one-quinoid structure amongst eight monomers, and it exhibits a transparent (solution) and white color (solid). Emaraldine has a two-quinoid structure among eight monomers, and it appears as a green (doped) and blue color (undoped). Nigraniline has a three-quinoid structure amongst eight monomer rings, and it appears as a dark blue color. The completely oxidized form is pernigraniline with alternate quinoid and benzenoid structures, and it is deep violet (solution) and black in color (solid) [108,109]. Amongst the above forms, PANI emeraldine salt can satisfy the high electrical conductivity ranging from 10^−3^ to 10^2^ S cm^−1^, depending on the conjugation length, dopant type, and dopant concentration [110].

PANI conductivity enhancements were related to the formation of nanofiber morphological transition, and such non-conductive to conductive transitions were achieved with the help of a variety of dopants. This transition ability to switch between non-conductive and conductive forms promoting PANI to sense volatile organic compounds because of the acidic/basic and reducing/oxidizing nature of the studied analytes [111]. Similar to other conjugated polymers, pristine PANI remains limited because of its poor solubility and processing parameters. The reason for its poor solubility and processability can be linked to the presence of an extremely rigid PANI molecular backbone and strongly conjugated π electron system. Solvent solubility and processability can be solved with protonation or by forming the substituted derivatives [112].

Conventional PANI synthesis through chemical oxidative polymerization aids in achieving the conductivity, but failed to exhibit the facile processing conditions. On the other hand, VPP technique serves as influential in solving the solubility issues, and it exerts good solubility, even at room temperature, high thermal stability, and better crystalline features. Unfortunately, the conductivity is compromised with VPP process [108]. In short, nanostructured PANI morphology is achieved by aniline vapor through an aqueous acidic solution of oxidizing agent.

Unlike conventional nucleation process utilizing the conventional inorganic protonic acids, such as hydrochloric acid [113], sulfuric acid [114], and phosphoric acid [115], the dopants further evolved into organic acids [115], binary dopants [116], etc.

For instance, Emanuelsson et al. successfully employed the organic polyacid doped PANI membranes with balanced porosity and stability, giving rise to better organic solvent nanofiltration process [117]. Dodecylbenzene sulfonic acid doped PANI formed onto the bacterial cellulose membrane with added control over the oxidizing agent and dopants gave rise to better electrical conductivity [118]. Qi et al. further studied the same dopant on the surface of viscose fiber in eco-friendly solvents water and ethanol. Subsequent optimization of solvent ratio, monomer, oxidant, and dopant eventually led to the good durable conductive fibers capable of enduring its functionality even after washing [119].

Traditional strain sensors are mostly made of metals or rigid materials, and their strain is very limited (<5%). At the present stage, strain sensors have undergone explosive growth in recent years, and its working range extends over 800% due to the utilization of novel polymeric elastomers. Numerous developments have been made with metallic nanostructures, including morphological hybrids, welding, architectural gradients, and aspect ratio modifications. Although achieving metal-like conductivity was satisfactory, other important factors such as loading control, corrosion, compatibility, and toxicity factors seriously impedes the sensor evolution. The conducting polymer’s conductivity falls within the range between metal and semiconductor, and its tunability remains facile with various dopants. Dopants developed with various acids, insulating polymers, and other conductive hybrids. Following this, Irfan et al. in situ polymerized PANI dodecyl benzene sulfonic acid DBSA particles within thermoplastic PU (TPU) matrices, and their study reveals that excessive concentration of aniline monomer results in polymerization difficulties because of the oxidant diffusivity controlled by the TPU molecular chains [120].

Photoplethysmography (PPG) sensors record the vascular pulsation by measuring the intensity of light transmitted through or reflected by a tissue. It is widely used for pulse wave monitoring, but the accuracy of the sensor is affected by the contact between the PPG probes and the tissue. Ultrasound pulse sensors work on the basis of wave penetrating ability provided by the ultrasonic wave, but the sensory device remains bulky and rigid, and is unsuitable for patching and wearability. Kang et al. fabricated strain sensors PANI-bonded silicone elastomer and doped laminated to achieve robustness and linearity. Interestingly, dopant solution concentration effectively alters the wrinkling patterns and the wavelength of the patterns. The additional factor of thickness with its relation to sensitivity and linearity was clearly highlighted [121].

Conductive hydrogels have gained interest for its self-healability, and such materials seriously suffers from low sensitivity, slow response and relaxation times, and lack of recoverability. Moreover, hydrogel networks are prone to failures even when applying nominal forces. To avoid those mechanical failures, it is addressed with strong chemical covalent bonds and an abundance of weak physical cross-links, i.e., hydrogen bond and electrostatic interactions. Wei et al. developed interpenetrating PANI moieties and polyvinyl alcohol (PVA) chains cross-linked by glutaraldehyde. Phytic acid effectively served a dual role, i.e., acid dopant for in situ polymerization to form PANI and the catalyst for the cross-linking reaction of PVA. Strong chemical covalent bonds existing in PVA/phytic acid and PVA/glutaraldehyde, in addition to the weak physical crosslinks, i.e., hydrogen bonding and electrostatic interactions among PANI conductive networks, granting good toughness (2.5 MJ·m^−3^), and low elastic modulus (1.0 kPa), promoting them as wearable sensors [122]. Stretchability along with the reprocess ability and reshape ability achieved with reversible Diels–Alder-networked PU materials and the conductive PANI networks established with phytic acid dopants were in-situ polymerized. The as-designed resistive sensors exhibit better sensitivity, and the working range extended up to 100%. Apart from 100% stretchability, the durability cycles were performed for 25% strain to study the reliability of the strain sensors [123]. Another strategy to improve the sensitivity is by forming the piezoresistive sensors’ designs with micro-cracks, and the freestanding 3D PANI sponge employs a simple electropolymerization technique of Ni foam as a sacrificing template (Figure 5a). The generated 3D spongy sensors can differentiate the different heartbeat rates, revealing the comparable resistive signal peak counts and intensity [124].

Microstructural architectures keep progressing towards better sensing characters, as they may impose electrical changes even under subtle strain and pressures. Following this, tile like MXene/PANI nanocomposites sensing layer, developed by spreading MXene and PANI layer on to the elastomeric rubber. Tile structures favored the micro crack formation, propagation, and reversible overlaps over the nanosheets. The tile-structured strain sensor collectively executes wide sensing range (up to 80% strain) human motion with an ultralow detection limit (0.1538% strain), high sensitivity (2369), excellent reproducibility, and stability [125]. The integration of PANI nanowires into 3D graphene sponge manifests the pressure sensor with plenty of conductive network formations and offers balanced-sensing attributes such as good sensitivity and sensing range. Additionally, the cyclability records good values due to the ordered robust microstructures with a higher elastic modulus [126].

In the past, the fabrication of sensory devices remains technically challenging, due to the laborious, synthetic, and fabrication complexities. Assembling the strain sensor sandwich structures can cause easy deformation and create good electrical responses. To achieve better sensitivity, sandwich-structured sensors are composed of graphene meshes and paper-based PANI. The resultant paper-based flexible sensor harvests sensitivity of 800 for 7% strain. Still, the sensing range is very limited, as with conventional rigid sensors [127]. Similarly, paper-drawn graphitic layers serve as platform to create PANI nanofibers via an in-situ anodic electrochemical polymerization, which functioned effectively in supercapacitors and elbow strain-sensing applications [128].

Strain sensor fabrication supported by the blends of nitrile butadiene rubber and PANI with DBSA dopant-assisted in situ polymerized thin films. Alignments of PANI microtubes over 20% strain gives an interesting trend of increased conductivity, whereas, on higher strain percentage, PANI microtubes with airy voids experience disruptive conductive pathways. Thermal stability acquired with the as-prepared composites is significantly better due to the enhanced interactions between the polymeric phases, thereby restricting the chain motions even at higher temperatures [129]. Additional efforts made by Huang et al. considering the lower strain working range limitations. Synchronizing the graphene nanoparticles (GNP) with PANI particles, taking advantage of its smaller dimensions, offers facile accommodation of PANI interstitial particles among GNPs. The resulting composite GNP/PANI/silicon rubber strain sensors worked with better sensitivity of 67.3 (0 to 40% strain), which is four times better than GNP/silicon rubber. The working range and sensitivity increments is related to the PANI inter-connective synergistic conductive networks [130].

PANI aggregates easily in aqueous solution due to its poor dispersibility. On the other hand, elastomer natural rubber is highly viscous because of its higher molecular weight. Such characters harm the better dispersion, and it is challenging to achieve conductive composite thin films. Meanwhile, the poor dispersion and randomly distributed PANI agglomerates in elastomer matrix critically degrades the mechanical strength and electrochemical performance of hybrid elastomers. Desired stable suspension made viable through electrostatic charges existing in the cellulose nanofibers and additional amphiphilic surfaces promotes it as an ideal dispersant and template to hold PANI in an aqueous medium. To combat the poor dispersion issue, cellulose nanofiber templates were surface functionalized with PANI polymer (Figure 5b), thereby uniform dispersion was achieved via a latex co-coagulation process [131].

The hydrophobicity of the nanomaterials leads to excessive agglomerations, which are most common in hydrogel, and such aggregates might potentially harm the strain sensors upon excessive strain application. Implementation of a synergistic combination of hydrophobic association and nanocomposite-based strain sensor fabrication smoothens the hydrophobicity problems. In particular, PANI shell structure offers hydrophobic association between copolymer and the core, thereby affirming the uniform dispersion due to the physical crosslinking aspects [132]. Despite achieving body motion, strain sensing, and pressure sensing, sensors on board were capable of monitoring strain along with different parameters including pressure, temperature, and volatile organic compounds, to perform multisensory tasks. However, it is of greater challenge to stabilize a performance under lateral strains. Exposure of PANI thin films towards hydrochloric acid dopants for a few seconds presents a visual color change from blue to green, affirming the conversion of pernigraniline base to an emeraldine salt. Along with the color changes, it exerts micro wrinkle structures on the top surface with good uniformity [133]. Although the thin films and other microstructures are effective in terms of sensitivity, working range, etc, the directionality, hysteresis, wearable comfort, air permeability, active site modifications nevertheless remain challenging in thin film micro-structured sensors. Nanofibers are well known for their facile synthesis, processing, scalability, and their functionalities promote them as sensors, photocatalyst, LEDs, Lasers, and wearable electronics. In accordance, nanofibers with nano-branched coaxial geometry formed by utilizing electrospinning technique. PANI nanobranches deposited over the surface of polyvinylidene fluoride (PVDF) fibers on straining can deform and break, ultimately granting the resistive electrical responses [134]. Nanofibers are well known to produce beta phase shifts during the electrospinning process as it provides the stretching forces on passing the voltage. PANI loading on to the PVDF nanofibers with different flow rates and the ability to form the electroactive beta phase crystallization were evolved simultaneously [135].

Textile-based multimodal sensors should be capable of determining the strains and other olfactory stimuli with a single framework design. Differentiating the stimuli signals can be of beneficial use, and the stimuli responsivity mechanism must vary to produce distinct resistive signals. Following the requirements, a pressure sensor capable of sensing 30 to 300 kPa and ammonia sensing is developed, wherein the pressure sensor causes resistance to lower and ammonia withdraws protons, causing resistance to hike, apparently. Moreover, the high sensitivity and fast response of the PANI textile-based sensor to ammonia is related to the high surface-to-volume ratio of PANI, promoting more active sites available for gaseous molecules’ adsorption [136].

Pressure sensors based on conductive polymers follow the architectural differences, and the strain application varies in terms of compression, rather than tensile deformation. Apart from conductivity changes adopted by the doping process of PANI with DBSA, it also provides a better pressure sensory response. The ratio of dopants to monomer optimization on forming the pressure-sensitive pellets along with good cycling stability. Moreover, the pressure applied to form the pressure-sensing pellets has a stronger correlation in achieving the better pressure sensitivity in correspondence to the conductivity gradients [137]. Multiwalled, nanotubes doped PANI excels with good conductivity responses in response to pressure application. Composite offers continuous channel effect where nanotubes form a bridge with PANI matrices. The pellets sensory response appears vivid because of the significant void coverages that induces instant conductivity changes. Furthermore, the conductivity variations are obviously higher for the composite conductive matrices than the pristine substrates [138].

Conventional gold nanowires present an excellent aspect ratio, although the conductivity achieved with PANI doping through a simple pen brush painting method on curved substrates. The advantage of using PANI microparticles on to the nanowires is sea-island formation, where elastic nanowires can absorb forces rapidly and stretch with better reversible characters, leaving the PANI microparticles undisturbed. A considerable degree of stretching enhancements achieved with a curved tattoo made of varying bend radii [139]. The stretchability improved with curved structures allows the fabrication of directional strain sensors, and it is of greater significance to design directionally responsive multidirectional sensors. The utilization of magnetic-controlled nanoparticles with nano morphological tunings facilitates the formation of aligned silver nanostructures- PANI composite films. Interestingly, the resistive response contrastively provides the sensory outputs with respect to a parallel and perpendicular mode of sensing [140]. Recently, commercial sponges were carbonized into airy 3D carbon foams with thermal treatments. Three-dimensional carbon foams coated with reduced graphene oxide nanosheets and PANI nanorods paved the way for substantial improvements in energy storage and pressure sensing with good sensitivity. Pressure sensing sensitivity, achieved with continuous stress monitoring, was made plausible with such ternary composites, eventually providing a better path for the rise in current signal response [141].

The electrostatic interaction between the positively charged PANI and negatively charged reduced graphene oxide nanosheets can promote the π–π stacking and hydrogen bonding between the functionalities among the two molecules, and can positively contribute to the pressure sensing device fabrication. The idea of creating this was implemented through a reduced graphene oxide coating and in-situ polymerization of PANI nanowires onto the spongy substrates. The as-designed sensor possesses tunable sensitivity (0.042 to 0.152 kPa^−1^), a wide working range (0–27 kPa) and a high current output (∼300 μA at 1 V). Collective superior performance attributable to the conformal coating of PANI-sensing elements over the pristine sponge, and low elastic modulus coupled with a high compressibility favoring the higher current response [60]. Although chemical interactions played a major role in achieving tunable sensitivity and structural and geometrical engineering, the sensory active layers promisingly contribute in heightening the sensitivity and extending the sensor working range. Fracture mechanics involvement with spongy structures’ improved sensitivity and linearity of the pressure sensors and the detection limit is 4 Pa. Permanent micro-cracks substantially improved the resistive response even under the lower strain application, and subsequent higher strains can give rise to elevated resistive signals. From the synergistic cracked and 3D structures, Zheng et al. verified the pressure and strain sensing versatility and linearity [124]. Very recently, the wide range linear responsive pressure sensor was designed with a combinatorial approach, composed of a hollow structure and micro-protrusions. PANI composite with commercial polymer, wider operational range, about 0.05 to 60 kPa. Structural construction parameters crucially modifies the pressure sensor performance, yet another wearable pressure sensor framework consisting of eco-friendly cellulose paper with a PANI fabricated with easy dip-coating method. Interestingly, PANI cellulose paper stacking optimization critically outperformed the conventional single sensor sensitivity and working range. Increments in sensitivity (2.23 kPa^−1^) and pressure sensor working range (5–22 kPa) can be correlated with the porous structures and interfacial air gaps [142].

In the past, in situ polymerization methodology is widely adopted in forming PANI conductive layers over various substrates. However, the control of loading and morphological characters is unsatisfactory. In some 3D structural sensors, non-uniform distribution of PANI microstructures is commonly encountered due to the diffusion and polymerization process complications. Wang et al. improved the dispersion of PANI over the gel structures by utilizing the new solution assembly method. Pre-synthesized PANI assembled with 3D polyvinyl alcohol chains allowed to gelate and crystallize by taking advantage of hydrogen bond interactions between PANI and PVA [143]. Natural biomass bacterial cellulose coupled with chitosan and in situ polymerization to form the conductive composite, where bacterial cellulose and chitosan are natural and biodegradable, and PANI exerts piezoresistive sensory response. The resultant aerogel obtained by a freeze-drying method produced a compressible, resistive response with low pressure detection (32 Pa) and good sensitivity (1.41 kPa^−1^). Partially ruptured composite in the first few bending/releasing cycles contributed detrimental electrical hysteresis behavior, and, interestingly, composites achieved saturation after several bending cycles, which indicates the great reversibility of composite [144].

Challenges such as freestanding and avoiding the sacrificial layers to fabricate the continuous thin films makes the sensory device fabrication demanding in terms of scalability [133]. Several works based on wearable devices have also exerted the contradiction of achieving the strength and the toughness of the devices because of mechanical modulus’ mismatch experience between the skin and wearable devices [145]. Efforts pertaining to solve the mechanical, electrical stability synchronously can make significant differences in creating the highly durable devices with lowered hysteresis.

## 8. Polythiophene Sensors

Among the conducting polymers, polythiophene and its derivatives are the most investigated for designing the spectrum of optoelectronic applications such as solar cell, LEDs, field effect transistors, memory devices. The most contributing factor to the successful implementation of these polymers are polythiophene optical property tunability, which can be easily achieved by pre-functionalization of monomer. Polythiophene derivatives exhibits solution processability, which renders the thin film fabrication process in attaining the good optical properties [146]. In particular, the solubility and processability of polythiophene derivatives capable of facilely fine-tuning the design, molecular weight, intrachain and interchain *π*-overlap. Several breakthroughs have made the *π*-architecture and the extent of conductivity enhancements and solution processability significantly depend on the synthetic design and reaction pathways. The regioregular head-to-tail-coupled polythiophene is a leading example used to understand the importance of the synthesis of materials [147]. Numerous techniques have been evolved and are crucial in achieving the proper design and strategic synthesis to prepare beneficial *π*-conjugated polymers. In general, thiophene derivatives polymerization were preferentially carried out via chemical and electrochemical methods [146,148]. In the past, the major breakthrough was the preparation of beta-substituted thiophene monomers formation, and its successive polymerization to form solution processable polythiophene derivatives. Intriguing results achieved in terms of conductivity enhancements after p-doping and such conductivities inlaid the foundation for polythiophene based novel materials and various potential applications [149,150].

Performance enhancements of polythiophene derivatives developed via an in-situ polymerization of conjugated polymer poly [(thiophene-2,5-diyl)-*co*-(benzylidene)], which exerts non-covalent binding on graphite and graphene oxide sheets, was considered to evaluate the combinatorial effect of structure and doping towards cyclic stability [151]. Intriguing anomalous behavior of less ordered, i.e., nearly amorphous conjugated polymers towards achieving high charge transport characteristics utilizing the electrochemical polymerization process.

Among the aliphatic polyesters, the hyper branched polyesters based on 2,2-bis(methylol)propionic acid, have triggered great interest, mainly due to their highly branched structure, large number of functional groups, and unique physicochemical properties, posing as a significant advantage for industrial and biomedical applications. Novel three-dimensional (3D), conducting, biocompatibility, and a porous scaffold composed of hyperbranched aliphatic polyester, polythiophene, and poly(ε-caprolactone) are used in tissue engineering applications [152]. Polythiophene has been widely studied, and its conductive natures are mainly dependent on the molecular structure modification, nanostructure control and composite. Polythiophene monomers are susceptible to grafting, and are efficient in forming the respective polymers, such as poly (3-methylthiophene), poly(3-hexylthiophene) (P3HT), and PEDOT. It is evident that P3HT undergoes recrystallization in the presence of conventional non-crystalline polymers such as poly(methyl methacrylate) (PMMA) and polystyrene (PS) to form nanofiber composite films [153]. Shimomura et al. carried out the formation of transparent conductive nanofibrillar P3HT by polymeric composite formed with PMMA polymer and oxidant solution AuCl_3_ [154]. Flexible conductive film possessing high conductivity achieved through optimal ratio tuning of acetonitrile and boronic agent [155].

Doping agents on polymerization of conjugated polymers brings critical advantages such as conductivity, solvent solubility, and good dispersion. Traditional dopants were inorganic salts, and their oxidizing power is better than organic dopants. However, the resultant polymer exhibits poor solubility in the solvents, thereby restricting the applicability. Following this, Bronsted acid and Lewis acid dopants comparative studies revealed that the Bronsted acid doped P3HT lead to the delocalized length increase because of enhanced crystallinity and backbone planarization contributions [156]. As an alternative approach, molecular structure and crystallinity fine-tunements of the polythiophene remains crucial in glorifying the high charge carrier mobility of the transistors. Conductivity achievements and the fabrication methodology evolution favors the formation of conductive layers over the dielectric substrates. Following this, conducting polymers functioned as the seeding layer to activate the metal electroplating process, which has previously been challenging to achieve with the insulating substrate. Interestingly, polythiophene played the role of an activating layer in promoting the smooth conductive nickel nanoparticles’ formation on the insulating matrix [157]. Polythiophene functionalized multiwalled carbon nanotube binary composites stabilization was achieved with sodium bis(2-ethylhexyl) sulfosuccinate micelles prepared by oxidative polymerization. Binary composite, along with tangled silver nanoparticles’ embedment, led to the development of ternary nanocomposites. The as-prepared ternary composites excel with better electrical conductivity (80.76 S/cm), and such conductivity is ascribed to the effective charge transport through the polythiophene layer, which acts as conductive bridge between multi-walled carbon nanotube (MWCNT) and silver nanoparticles [158]. Betterment in electrical conductivity can give rise to better excellence in sensory applications. Analytes respond with electrical conductivity/resistance variations, as conducting polymer moieties can either function as electron acceptors or donors [159].

Regardless of conductivity, sensitivity, and processability improvements, another important criterion for the industrialization is their mechanical robustness, as that may reduce the pace for the polymer-based electronics development. Since then, the mechanical property remains a challenging task for the polythiophene products, such as P3HT [160], which serves as a severe impediment for the growth of organic electronics and their industrial applications. Semiconducting polymers generally exhibit poor stretchability because of the high crystallinity presented by their rigid backbones and strong π–π interactions, therefore the anticipation towards the evolution of intrinsically stretchable semiconducting polymers is growing gradually [161]. The flexibility and bendability of P3HT are not sufficient to tolerate sequential times of multiple bending processes. Such mechanical instability causes fluctuations in the electrical properties of P3HT-based flexible devices [162]. To resolve the issue, high molecular weight and highly regioregular poly(3-substituted thiophene) with disiloxane moieties in the side chains developed mechanical robustness. Furthermore, the investigation of molecular structure and physical properties of the substituted thiophene affirms the excellence in flexible mechanical properties. From the past, it is well known that the influence of side chains on the chain mobility and glass transition temperature (*T*_g_) (or the flexibility) of the as-prepared substituted polymers is influential. In particular, a bulkier side chain substituent tends to cause a greater stiffening (higher *T*_g_) by means of bond rotation restrictions [160]. Bond rotational restrictions can be achieved with a branched side chain, i.e., P3HT isomer, poly(3-2-methylpentylthiophene), bearing methyl-branched side chains, rather than utilizing conventional linear chain polymer. Methyl group on the side chain selection aids in restricting the rotation of the adjacent C–C bonds, and, because of it, side chain turns more rigid in comparison to the linear one on P3HT. Such chain motions are restricted with branched side chains, and aid in decreasing the crystallization-induced phase separation, which acts a crucial factor in improving the polymer solar cells’ stability [163].

Another method to improve the air stability is to employ the ester-substituted side chains with more free space. This ester side chain substitution biaxially extends the conjugation to improve the mobility and stretchability attributed to the enhanced amorphous structure with reinforcements [164]. Interestingly, the crystallite orientation during the drawing process is mandatory in deciding the electrical conductivity. Furthermore, polythiophene disiloxane-substituted derivatives possess a specific crystallite orientation, which remains stable up to a tensile strain of about 140% [165]. Control over the crystallization modes of conjugated BCPs based on poly(3-dodecylthiophene) and poly(2-vinylpyridine) can be of achieved with P3DDT regioregularity, due to the melting temperature control and crystallization rates of P3DDT. Confinements achieved with low recovery ratios, crystallization at temperatures near or below the *T*_g_ of P2VP, and such crystal growth confined by the glassy cylindrical or lamellar BCP structure [166]. Stretchable active channel matrices composed of polystyrene-*block*-poly(ethylene-co-butylene)-*block*-polystyrene (SEBS) and P3HT composite solution obtained with spin casting process. The successful formation of the in situ phase separation of the P3HT nanofibrils to the surface of the rubber matrix, assembly of nanofibrils into wide bundles, network formation of the bundles, and indentation of the bundles on the rubber surface functioning are key factors [167].

Significant progresses is made with block copolymers consisting of conjugated polymer and rubbery soft polymer blocks, as they offer phase separation of block copolymers, which controls the electronic functions significantly. Following this, Chen et al. utilized a similar block copolymer strategy designed with a rod–coil donor acceptor polyfluorene-block-poly(pendent isoindigo) for stretchable memory applications, and the resultant devices worked reliably under stretchable conditions [168]. From this perspective, click reaction-processed poly(3-hexylthiophene)-*block*-poly(butyl acrylate) rod–coil diblock copolymers (P3HT-*b*-PBA) was made via alkynyl-functionalized P3HT and azido-terminated PBA homopolymers. Moreover, the block ratios and their effect on mechanical and morphological aspects were studied to employ them as stretchable FET devices [169]. On a similar basis, Higashihara et al. reported a coupled synthetic path using Kumada–Tamo catalyst transfer polycondensation and living carbocationic polymerization methods to prepare ABA triblock copolymer (P3HT-*b*-Polyiosobutylene-*b*-P3HT). Polyisobutylene block selection was made due to polymerization reaction free of heteroatom interactions, making morphologically tunable, deformable, and elastomeric polymeric resultant matrices [170].

In recent years, the utilities of electronic portable devices (e-books, smartphones, and tablets) has reached its peak due to COVID-19 outspread, resulting in the majority of work being carried out from a home environment through online resources and platforms. In the future, scientific advancement can reach greater heights by a series of technological hotspots, including the internet of things, higher-potential consumer electronics, including portable electronic devices, such as patchable and wearable sensors, and biomedical and output displays. Excessive mechanical deformation under flex and stretch conditions may potentially interrupt the electrical performance of these electronic devices due to stress concentration, distribution, and relaxation factors due to the disconnection between the conductive network paths. In recent decades, there have been trends of fabrication approaches with green chemistry aspects exploiting natural monomers or blocks for constructing the active components in various optoelectronics. For instance, Kuo et al. studied the morphological and charge transport mobility effects of solvents on the P3HT and poly (lactic acid) blends. Solvent compatibility differences among dichloromethane and chloroform led to the formation of well-defined, self-assembled P3HT nanowires [171]. Naturally-derived collagen hydrolysate can be facilely processed in an aqueous medium for further optimization of crystallization, significantly playing its major role in achieving better memory characteristics [172]. Exploration of polythiophene and its composite has not been successful in improving the strain sensing, as its electrical conductivity factors are not comparatively better than polypyrrole and polyaniline conducting polymers. In contrast to other conducting polymers, PEDOT shows better conductivity, higher transparency, and possesses great environmental stability. However, this material is insoluble in water or organic solvents and is thus difficult to apply using casting or spin-coating techniques. To overcome the solubility and processability issues, Bayer prepared a graft copolymer of PEDOT with poly(4-styrenesulfonate) (PSS) as an aqueous colloidal suspension. Facile processing and solubility conditions have made PEDOT PSS significant in the fabrication of solar cells, LEDs, sensors, etc. The successful polythiophene derivative PEDOT and its sensory performance have been presented in an upcoming section.

## 9. PEDOT Sensors

Thiophene ring substitution with 3 and 4 positions affords Poly(3,4-ethylenedioxythiophene) (PEDOT) with a considerably better corrosive resistance towards ambient and humid conditions. Pristine PEDOT or doped PEDOT structures exhibit poor solubility, and this leads to the evolution of Poly(3,4-ethylenedioxythiophene):poly(styrenesulfonate) (PEDOT:PSS), wherein PEDOT is positively charged and PSS is a negative charged water-soluble component. PSS plays major functions, such as acting as a counter component to stabilize PEDOT dispersion and maintaining its integrity in the aqueous medium [173]. PEDOT:PSS falls under a category of polyelectrolyte complex, and it is prepared by the oxidative polymerization of EDOT in the presence of PSS. In solution, negatively charged PSS higher molecular weight chains are tightly surrounded by relatively shorter oligomers of PEDOT. Moreover, PEDOT:PSS, serves as an excellent hole transport layer in organic solar cells and as a hole injection layer in LEDs. Commercially available formulations differ primarily on the basis of the ratio of PEDOT and PSS. The formulation Clevios PH1000 type possesses a weight ratio of 1:2.5, and the Al4083 type possesses a weight ratio of 1:6. For instance, Dupont et al. assessed the moisture-assisted decohesion process in PEDOT PSS thin films, and the decohesion mechanism could be plausibly connected to the hydrogen bond predominance in bonding the individual PEDOT:PSS grains within the layer [174]. Veeramuthu et al. effectively nullified the luminescence quenching contributions offered by the PEDOT PSS hole injection layer with the use of an interface-assisted grain control process, which resulted in the synchronous efficiency and stability of LEDs [18].

Tertiary composites, made of PEDOT:PSS/dimethyl sulfoxide (DMSO)/zonyl, their detectable response at 20% strain and elastic deformation. Zonyl component supports the formation of conductive networks, as many of the previous research studies have attained conductivity with mere solvent treatments [175]. Strain sensors fabrication with PEDOT:PSS ink and silver nanoparticle inks were facilely made using inkjet printing. The resulting sensors’ sensitivity of silver ink are about three times better than PEDOT:PSS, and the sensitivity differences can be correlated to the dimensional variation of the conductive ink. Cyclic strain performance altered with respect to cycle numbers, however their relation remains unexplored [176]. PEDOT:PSS composites made of polyethylene oxide (PEO) and amphiphilic fluorosurfactant zonyl exhibit better conductivity with a balanced higher transparency of 95% at a 550 nm wavelength. Amphiphilic zonyl addition delays the formation of cracks on straining, which is evidenced with zonyl concentration optimization. In precise terms, composite electrodes with 5 wt% zonyl showed a lowered resistive response upon the 120% strain. Furthermore, upon reducing the zonyl concentration to 0.5 wt%, R/R_0_ increases 360-fold, which is influenced by the detrimental micro-cracks [177].

Buschbaum et al. studied the effect of humidity-induced changes of PEDOT:PSS electrodes with the help of in situ time-of-flight neutron reflectivity (TOF-NR) measurements under relatively high humidity conditions. The role of solvent treatment, the addition lf zonyl, and a post-treatment of PEDOT:PSS films with ethylene glycol were rationally investigated with respect to the swelling ratio and water uptake. PSS content in PEDOT’s composition works as a prime factor in improving the water processability. However, excessive PSS component will eventually degrade the device’s performance, and it has a major influence in charge transport properties [178]. Moreover, the hygroscopic nature of the PSS components eases the water molecule absorption, causing degraded electrical conductivity values. Water intake by PEDOT:PSS threatens the organic optoelectronic device fabrication, as it clearly disrupts the structural integrity and causes mechanical stress over the functional layers. In situ TOF-NR measurements under high humidity conditions enables one to probe the swelling kinetics. Swelling kinetics observed with in situ TOF-NR enables the comparison of humid and dry PEDOT:PSS film stability towards water intake. PEDOT:PSS swelling and water intake was significantly curtailed by the combinatorial approach of the zonyl additive and ethylene glycol crosslinking treatment [179].

For compressive type strain sensor or pressure sensors, it is desirable to achieve higher resilience with negligible mechanical hysteresis. Polyimide nanofibrous aerogels are promising in achieving higher resilience towards compressive, folding, and torsional strains. Three-dimensional nanofibrous porous aerogels welding is done with various organic solvents and with optimized time and temperatures. In particular, the champion mechanically resilient welded strain sensor was achieved with a 60 °C DMF solvent vapor treatment [180]. Improvements in durability are challenging and have posed consistent challenges in the past, which are addressed via morphological engineering, microstructure patterning, and composite reinforcements. Following this trend, sandwich-type structured strain sensors are appealing, with good sensing characters. A highly conductive PEDOT:PSS film is embedded between a PDMS elastomer and a PEDOT:PSS film doped with PVA and Zonyl. The plastic strain sensor demonstrates high sensitivity, strain sensitive region of 5–30%, and good durability. Better durability and recoverability in electrical response have an inter-relation with the stable structural integrity and strong interconnected conductive networks [181]. Another sandwich type of strain sensors made of PEDOT:PSS/AgNWs’ conductive composite successfully transferred into elastomer PDMS using a nitric acid-assisted transfer printing method. High yield transfer printing made possible via weak Van der Waals interactions between the PEDOT:PSS substrate upon acid treatment. Robust sandwich-type integration offers stretchable and reliable performances with good sensing range and sensitivity [182].

Aerogel direct immersion on to the ethylene glycol solution causes mechanical instability to the PEDOT PSS/cellulose nanofibrils’ aerogel composite; Zhou et al. proposed ethylene glycol solvent vapor annealing, and further thermal annealing process led to the highly conductive aerogel matrix. Interactions between the cellulose nanofibrils carboxylate/carboxyl groups with PSS aids in transforming the benzoid to be electrically conductive quinoid structures. The resultant conductive aerogel strain sensors that are highly stretchable (100% strain) and sensitive (GF = 14.8) with high linearity [183]. PEDOT:PSS post-treatment with dichloroacetic acid removes excess PSS, facilitating conductivity enhancement, and a PSS-free PEDOT piezoresistive response can remain selective to strain rate, irrespective of humidity factors [184]. Sensitivity towards gaseous molecules and their sensory response significantly speeds the altered process by reducing the diameter of the electrospun nanofibers through a high-pressure, airflow-assisted electrospinning technique [185]. Carbon-fabric-based strain sensors failed to execute high sensitivity and large stretchability simultaneously, thereby limiting its applicability towards human bodily motion sensors. A comparison of 2D and 3D structured strain sensors was designed to pronounce the achievement of good sensitivity and sensing range. The comparative results suggesting that the significance of the 3D structured strain sensor attained a broad sensing range from 0 to 180% [186]. Another step forward made to balance the sensitivity and sensing range of conductive composite PEDOT-based strain sensors via deploying the unique microstructures and stronger adhesion between PEDOT:PSS and one-dimensional (1D) AgNWs. Near field electrospinning technique supports the fabrication of groove structures, and the additional hybridization of PEDOT:PSS facilitates the adhesion to the elastomeric substrate and multiscale electron transport path possibility on deforming it to greater extent [187].

The commercial glove surface penetration method was adopted to form carbon black gradient filler composition over the glove substrate. Gradient conductive filler concentration improved gradually in accordance to the immersion time factors. Typical gradient conductive networks afford a wide detection range of about 300%, with a better reproducible sensory response. Gradient structures allowed for better harvesting of hysteresis characteristics on straining by up to 100% strain [188].

Many of the previous studies limited to either strain or pressure detection. The sensor with simultaneous strain and pressure sensing attributes were rarely studied, and its sensitivity is limited. To overcome this limitation, hybrid electrode system made of PEDOT:PSS/single walled carbon nanotube (SWCNT) takes advantage of highly stretchable transparent e-skin and offers a dual conductivity–conversion mechanism to identify pressures at various stretch levels. The as-fabricated sensor can be stretched by up to 150%, with a GF of 21.5 at 0−100% strain. PEDOT:PSS merely exhibits a shorter working range, about 3% strain, with a drastic resistance hike up to the GΩ level. Such an ultralow working range can be correlated to the PEDOT:PSS film’s fragile nature, where the conductive paths were abandoned and isolated (Figure 6a) [189]. Very recently, biomimetic snake skin scale fabrication was made possible through bilayer micro-cracked PEDOT;PSS to achieve the wide range strain sensing with better resolution [190]. However, the dynamic range of graphene/palladium films remains unsatisfactory, with very limited strain (∼10% operation). The fragile nature of the resultant film degrades its films’ suitability with wearable applications. Highly plasticized PEDOT:PSS serving as an active conductive binder and improved interfaces providing promising pathways for electron conduction even on lower strain to higher strains [191]. The printing technique is easier to fabricate a continuous intelligent device with low cost features and complications. Screen-printed patterns influence the linewidths and patterns, and have considerable effects in sensing performances [192]. The majority of the strain sensory devices are concerned with developing sensitivity, working range, and reducing the electrical hysteresis, leaving behind the directional performance. Directional working sensitivity is critical in elucidating the sensor’s dynamic attributes. Directionality of the sensory performance explored with x and y-directions of the patterned and wavy micro-structured sensors. Wavy structures aid in accomplishing the wide sensing range of 120% along x-direction and y-directional sensory performance, and was considerably better due to the increased stress during the strain deformation process [193]. Top-down approach such as kirigami patterning works interestingly in maintaining the conductivity over the wide strain range by taking part in stress concentration and distribution over the patterned device structures [194].

The cellular porous materials tend to effectively accommodate both compressive and tensile strains, which is quite common with different human motions. Freeze-drying followed by chemical crosslinking with 4,4′-diazido-2,2′-stilbenedisulfonic acid disodium salt, which was added to PEDOT solution, followed by freeze-drying offers porous structures (Figure 6b). The as-prepared aerogel with chemical crosslinking imparts mechanical robustness, and provides structural stability in the presence of moisture or in an aqueous environment [195]. Glycerol solvent selectively solubilizes the PSS counter ion in PEDOT:PSS to improve the conductivity of the composite. Removing the insulating carrier PSS from the conductive PEDOT component can possibly shift the conductivity positively. A multifunctional organohydrogel was developed based on a poly (acrylic acid) (PAA) skeleton with poly (3,4-ethylenedioxythiophene): sulfonated lignin as the conductive material and a water/glycerol binary solvent as the dispersion medium. An added benefit of using glycerol is that it can provide an antifreezing property, which is better than conventional hydrogels. Because of the PAA shrinkage and deformation in the glycerol/water binary solvent, the resultant organohydrogels possess wrinkled structures. Organohydrogel electrodes can accurately sense and provide electromyographic (EMG) and electrocardiographic (ECG) signals with distinct P, Q, R, and S waves, which is in good agreement with commercial Ag/AgCl electrodes [196]. Generally, hydrogel sensors failed to exhibit good mechanical properties, and such mechanical failures usually occur due to the poor dispersion of conductive fillers. To solve this issue, composite hydrogels made of PEDOT:PSS fillers in the PVA hydrogels to form the semi-interpenetrating network structure. Under the strain application, covalent crosslinking maintains the original network configuration, and the sequential PVA chain movements, glutaraldehyde chain, and PEDOT:PSS chains designed for effective energy dissipation and transfer of stress from the PVA matrix to the PEDOT:PSS reinforcements signifies the better sensory performances [197].

Compared with the strain sensors based on highly conductive metal nanostructures or carbon nanomaterials, the strain sensors based on PEDOT:PSS exhibit combined advantages, including conformable attachment on skin, easy configurable, processing simplicity, fabrication, and biocompatibility. Furthermore, the trade-off relationship between high sensitivity and wide sensing range existing for majority of strain sensors remains common in PEDOT:PSS-based strain sensors. PEDOT:PSS sensing network develops multi conductive networks in the strain sensor, ensuring the existence of the conductive networks even at large strains and granting the sensor with a large sensing range. Porous structure of the PEDOT:PSS aerogel lead to its interlacing with the infiltrated PDMS substrate, thus withholding the sensing composite structures, firmly causing the better sensor response’s reliability and stability [198].

## 10. Nanofiber Sensors

Textile tends to possess greater potential, and it has revolutionized various functional components, as it provides good comfort and protection to the users. Research projects and industrial collaborations are bringing rapid growth, and it is anticipated that a variety of electronic components can be made out of textile basic component fibers. Major developments have been vividly visible in wearable electronics such as sensors, solar cells, batteries, supercapacitors, LEDs, energy generators, etc. Various methodologies have followed to fabricate fiber-drawing, templating, self-assembly, phase separation, centrifugal spinning, wet spinning, electrospinning, etc., [199,200,201,202]. Among which, the electrospinning technique stands unique in harvesting long continuous fibers with good dimensional tunability. Dimensional tunability includes various crucial parameters such as diameter, length, thickness, aligned, random, bead supported, net like, core–shell, hollow architectures, etc., [16,203]. Electrospinning works on the principle of electric field supported elongation of polymeric solution via a continuous extension from a metallic needle to the collector. In detail, the utilization of the electric field facilitates the formation of electric charges on the surface of the polymer solution. Because of the interaction of charges with the electric field, the polymeric solution extends from the metallic syringe tip towards the collector through the attainment of stable Taylor’s cone [204,205]. Electrospinning polymer solution’s viscosity, concentration, conductivity, molecular weight, dispersion, solubility, flow rate greatly influencing the formation of continuous fibers [29,206]. The obtained fibrous structure can be scalable and have a facile control with a wide range from the nanometer to micrometer scale [207,208].

The major complication in achieving the conducting polymer fibrous structures is their poor solubility in common solvents, which seriously restricts the electrospinning process. Another crucial issue was conducting polymer’s low molecular weight and high molecular chain rigidity, which complicates the electrospun nanofibers’ formation [209,210]. Oxidative polymerization of conducting polymers onto the copolymer matrices certainly works with the possibility of forming nanofibers [211]. Anionic surfactant dopant-assisted polymerization of pyrrole led to the precipitation of polypyrrole particle and blending of conductive particles along with PVDF polymeric carrier solution resulted in the successful formation of nanofibers [212]. Gradual developments have been made to tailor the conductive nanofiber diameters without employing surfactants, including an interesting attempt made to combine interfacial polymerization with a fast mixing technique, sacrificing the surfactant role. The effects of mixing speed, reaction time, temperature, and oxidant/monomer molar ratio influences the nanofiber size variations, thereby strengthening and weakening the electrical conductivity. In particular, nanofiber size declined nearly half of its original size by accelerating the mixing speed [213]. Tradeoffs exist between the processing solubility and conductivity enhancements of PANI nanostructures. One may balance the tradeoffs using a plant-derived sodium phytate, dopant-based synthetic methodology, which is cost-effective and environment friendly. Sodium phytate plays dual role as dopant and crosslinker by forming an interconnected net-like structured phosphorylated PANI [214]. Following this, polyvinylpyrrolidone (PVP) carrier polymer is compatible with conducting polymer other than polyethylene oxide to form the nanofibers. The electrospun pyrrole-PVP nanofibers collected directly on the coagulation bath consists of dopant solution., whereas the collection process and dopant immersion is done separately with a two-step process. Interestingly, the lowest average fiber diameter recorded for the electrospun samples was around 440 nm for a p-toluene sulfonic acid doped one-step processed nanofibers, whereas dual step-processed nanofibers exhibited an average fiber diameter of around 500 nm [215]. The low stretchable nature of PVDF resolved with an electric field-assisted twisting fabrication approach successfully develops the strain-responsive threads with a stretchability of 27% while keeping the core–shell structure of individual fibers. This alternative approach of twisting the fibers into threads significantly improves the elongation ranges [216].

Microfluidic method of forming core–shell fibers were designed with carbon nanotube (CNT)/PEDOT:PSS composite in natural rubber matrix. Outer shell coagulants diffuse into the core structure and stabilize the core–shell structured strain sensor fabrication. Long, uniform, stretchable, and composite microfibers were prepared in a single-step microfluidic device, subsequently allowing the mass production. The as-developed microfibrous composite strain sensors exhibited high stretchability accompanied by a high linearity of about 1000% [217]. The importance of PANI nanofibers lies in achieving electrical conductivity, rather than other PANI nanostructures, and the vital conductivity enhancement is about 10-folds for PANI nanofiber-grafted fabrics. Reversible protonation and deprotonation facilitated with nanofibrous structures renown’s the cyclability and practicality. Interestingly, the dopants concentration effects with the polymerization medium also portrays the significance in achieving better conductivity [218].

Lim et al. decorated PANI hairy nanostructures over the PU electrospun microfibers, and the results suggest the importance of nanofiber alignments in achieving higher sensitivity (40-fold increments) (Figure 7a) [219]. The prime importance of strain sensor fabrication relies on the conductivity and the conductivity achievements after crossing the critical point, i.e., percolation threshold. Minimum loading to complete the conducting network bridge among the insulating matrix is commonly termed as percolation threshold. On or above this percolation threshold, the carrier transport can be achieved via a tunneling or hopping mechanism to contribute to better conductivity. In several cases, the conductive fillers compromise its conductivity on blending with polymers to form nanofibers. Elastomeric fiber composites with high electrical conductivity at relatively low filler loading attain mechanical properties, finding its potential in strain-sensing applications. The choice of solvent selection to function as a coagulating bath in a wet spinning setup determines the conductive percolation threshold. Solvent mixture and the loading of conductive polymer PEDOT:PSS are critically important for controlling the electrical conductivity characteristics of the composite nanofibers [220]. Wet spinning of the conducting polymer PEDOT:PSS, often carried out through an extrusion followed by poor solvent coagulation bath immersion for the fiber formation. The as-spun fibers failed to excel with good conductivity, and, for the conductivity improvements, often utilize post-treatments. Post-treatments work on the basis of removing non-conductive PSS counterparts, thereby producing satisfactory conductivity. A recent study replaced the conventional coagulation solvent bath with a concentrated acid bath, thereby promoting the insulating PSS removal within seconds (Figure 7b) [221]. Rapid breakthrough in improving the conductivity through silver ion diffusion into the fluidic graphene oxide fiber during wet spinning coagulation process. The as-designed sensor possesses greater sensitivity and works up to 20% strain [222]. Although the conductivity and sensing range have been progressing, the recovery ratio for the wet-spun fibers remains at a lower strain%, and Gao et al. framed the strategy of forming tri-composite (PEDOT:PSS/PVA/ethylene glycol) sensors, showing a promising recovery up to 20% strain [223]. Ions and small molecules on wet-spinning, post-treatments were validated to be a good approach to enhance the original conductivity ascribed to the attenuation of Coulombic interaction among PEDOT and PSS counterparts. Even lower concentration of lithium salts-based coagulation processes can effectively improve the mechanical resilience/recovery behavior of the strain sensor [224]. A very recent study has revealed that the addition of lithium salts into a coagulation bath can improve the mechanical and electrical conductivity due to the tight and ordered internal structure developed with addition of lithium salts [225].

Electrospinning of the conductive polymers is made viable through the utilization of carrier polymers. Carrier polymer doped with oxidant solution promotes the mechanical integrity of the nanofibers. Vapor phase deposition of EDOT monomer can convert the oxidant nanofibers into the resulting conductive PEDOT fibers [226]. Similarly, iron (III) tosylate oxidant-integrated nanofibers are formed with polyacrylonitrile carrier polymer. Oxidant integrated fibers serving as a template to facilitate the growth of conductive polymer under nitrogen atmospheric conditions [227]. Small sized dopants such as sulfosalicylic acid and sodium benzenesulfonate are used to improve the interfacial bonding between PEDOT and bio-derived cellulose fibers. In particular, odium benzenesulfonate dopant acts as a surfactant to enhance the EDOT monomer solubility, thereby improving the crystalline characteristics of PEDOT [228].

Commercial fabric materials ease the fabrication of strain sensors. Thin film silicon rubber based strain sensors exert good sensitivity, with a good strain sensing resolution limit of 0.05% [229]. Low-cost sensors are designed with a Meyer rod-dip-coated sheath core-structured single fiber strain sensor made of graphite sheaths and silk fiber core, granting a good strain sensing range of up to 15% and a cyclic response [230]. Recently, cotton thread and a cotton strip-loaded PANI sensor were fabricated to quantify the efficiency in terms of sensitivity, response, and recovery speed. The underlying reason why the cotton fiber-based sensors improved sensory factors was because they relied on the higher surface area exposure [231]. Similarly, PANI/cotton conductive knitted fabric fabricated via in situ polymerization of aniline was excellent in terms of higher linearity, and provided a maximum sensitivity of 30. Still, the cyclic sensing range for cotton-knitted fabrics is less than 10%; i.e., on exceeding the 10% strain range, the structural relaxation and PANI conductive layer disruptions severely harmed the sensing repeatable characters [232]. Following this, pristine PANI and PANI/TiO_2_ hybrids formed on the polyester knitted fabrics. Pristine PANI-coated polyester fabric suffers from weaker abrasion resistance. PANI/TiO_2_ hybrid fabric’s abrasion resistance improved gradually with an increase in the TiO_2_ blend ratio. Average GF values of vertical strain were 21.71 and 20.26, respectively, for pristine and hybrid sensors, whereas the GF of the horizontal strain remained 2.25 and 2.06, suggesting the directional behavior of the fabricated sensors. The strain sensor directional response in terms of GF value reflects with connection and slippage of knit loops in which the strain was applied [233].

Nanofibers are comparatively better than commercial fibers, as they offers improved surface to volume ratio and improved sensitivity. Electrospinning processed with a PU/DMF spinning solution, followed by a conventional in situ chemical polymerization of aniline monomers on the surface of PU nanofibers to deposit a conductive PANI layer. Strain sensitivity, i.e., average strain GF values, could reach 17.15 with applied stretching deformation (0∼110%) [234]. The grafting of polythiophene onto poly(methyl methacrylate) was carried out using chemical and electrochemical oxidation copolymerization methods, with thiophene monomer being used to produce grafted copolymer. In addition, the solution of the chemically synthesized grafted copolymer, and carrier gelatin was electrospun to produce uniform and conductive nanofibers. Despite the lower electrical conductivity and electroactivity levels, the balance in terms of solubility and processability was achievable with graft polymerization techniques. The average nanofiber diameters and electrical conductivity of the grafted copolymer/gelatin composite electrospun nanofibers achieved was 70 ± 30 nm and 1.6 × 10^−3^ S cm^−1^, respectively [235]. Purposeful agglomerates of CNT along with microcrack PEDOT:PSS formed over the core material twisted PU fibers, wherein CNT agglomerates typically supports the formation of conductive bridges, even at large strains. Figure 7c displays the as-designed multi-filamentous sensor excelled with GF value (350) and wide-strain sensing range (0 to 150%) along with lowered hysteresis [236].

Fiber alignments can be of simple and effective methodology to harvest wearable strain sensors with high sensitivity and high stretchability. Interesting anisotropic sensing along the sensing directions governed by the aligned microstructure of the conductive networks are constructed by TPU fiber joints in the electrospun network and fish scale-like sensing layer on the fibers (Figure 7d). The as-prepared aligned structural fibrous membranes shines with broad sensing range (0–150% strain), high sensitivity (593), and stable resistance to humidity and temperature, qualifying the wearable strain sensor requirements [237]. Non-woven fabrics differed vividly from the woven fabrics, which has ordered warp and weft alignments, and non-woven fabrics well known for their air gaps among the fibrous networks. The cotton fiber surface formed with nanocluster networks of PANI nanofibers of 50 nm in diameter. The designed non-woven PANI sensors can distinguish the percussion wave, tidal wave, and dicrotic waves, which are typically present in the heartbeat of artery pulses. Moreover, the pressure sensors worked effectively with a sensitivity of 46.5 kPa^−1^ in a wide linear sensing range [238]. Similarly, sandwiching the conductive TPU electrospun fibrous network/MWCNTs and a polyimide (PI) sheet patterned with gold electrodes between two dielectric polymer layers. A sandwich type pressure sensor affords high sensitivity (2 kPa^−1^), pressure sensing range (10 kPa), along with the low hysteresis deviation (6%) [239]. More recently, Liang et al. fabricated all nanofibrous composed piezoresistive pressure and strain sensors with a higher sensitivity of about 71.07 kPa^−1^, and it also offered good clinical health monitoring with a better distinction among the atrial contraction, right ventricular contraction, atrial venous filling, atrium relaxation, and tricuspid valve opening [240]. The critical stage of washing fastness was achieved with the carboxyl functionalized CNT/TPU/Dopamine composite using ultrasonic deposition process. Washing fastness achievements can be attributable to the interactions among TPU and dopamine polymers [241]. Hydrophobic composites formed with silk fibroin/PANI composites and its anti-wetting property were ascertained with the porous structures, entrapping the bundles of air pockets [242]. The durable nature of the sensors can be governed by the conductive fillers’ stability towards air, moisture, and other ambient environmental conditions. Surface-treated polyimide fabric and subsequent PANI in situ polymerized in aqueous solution. Ag/PANI/polyimide fabric not only had excellent shielding efficiency, and remarkable electrical conductivity, presents outstanding anti-corrosion resistance, thermal stability, and fastness. PANI fiber plays a fiber–fiber adhesive role, and such a unique structure remains promising in achieving the credible features [243]. Another effort was to design an optimized conductive polymer polymerization process, which decides the formation of compact conductive layers on the textile surfaces. Compact conductive layers can eventually restrict the diffusion. Ag coated fibers, with additional PANI layers found to possess better corrosion resistance, and additional binding forces can potentially elevate the durability and lifetime of the sensory devices [244]. Due to the ambient conditions or sensor working conditions, the sensor might experience electrical hysteresis, and such hysteresis can give rise to erroneous results. Such interference from the environmental conditions was curbed effectively by forming the bark-like micro-structured conductive composites [245].

The processing and fabrication steps can still be made facile with other approaches such as dip coating, brush painting, and printing techniques to attain scalable and low cost wearable sensors. Control of fillers’ aggregation with a number of brush painting cycles can aid in the formation of low resistive textile sensors [246]. Indeed, wearable applications possess important characters, with the conductive textiles hybrids and patterning being used to accomplish good flexibility along with an endurance to bending, stretching, draping, and shearing limits. The conductive textiles prepared by the aforementioned techniques might pose some limitations/disadvantages in terms of fatigue and long-term cyclic operations. For instance, smart textiles designed from metal wires may lead to increased stiffness and thus reduce the elasticity. On the other hand, metal/carbon materials coated on the textiles can face failures and difficulties with crack generation, disconnect, and insufficient skin pliability and compatibility during the bending/stretching of the textiles. Therefore, endeavors are still needed to develop innovative materials and techniques in view of establishing a new territory on next-generation smart textiles.

## 11. Significance of Conducting Polymer Composites vs. Other Alternatives

The conducting polymers and its composites offer a broad spectrum of applicability, with lowered fabrication complications and cost factors. Although several breakthroughs have been achieved with carbon based nanostructures, ranging from carbon black, graphene, carbon nanotube, and carbon dots, there is still a much left to accomplish. Nevertheless, the emergence of carbon nanostructures remain limited due to their poor dispersion, processing of toxic conditions, and production cost and time factors. A variety of metallic nanostructures, alloys, and metallic oxides have been taking up the challenging tasks of reigning the wearable electronics. Due to the cytotoxicity and heavy metal leakage issues, liquid metal and various alloys have not been successful in forming human- and eco-friendly wearable sensors and electronics. Conducting polymers stays unique in establishing the relationship between the device performance and fabrication, due to their chemical stability, facile fabrication, low cost, biocompatible, and tunable morphology. Additionally, the conducting of experiments involving polymer composite has, in recent years, grown rapidly, and is anticipated to be the stardom of the optoelectronic industry. Conducting composites serve promisingly in attaining the adhesion and durable nature of forming the conductive networks over the diverse substrates. In particular, this review highlighted the evolution and importance of conducting polymer and its composites in strain and pressure sensing applications. Conducting polymers and its composites can possibly extend its arms towards energy production, energy storage, internet of things, artificial intelligence, soft robots, artificial prosthetics, in the near future.

## 12. Conclusions and Future Perspectives

From the above progress, it is perceptible that strain and pressure sensing studies have significantly grown, which has led to obvious developments in terms of e-skin, soft-robots, actuators, synaptic devices, and smart textiles. Self-powered energy-generating systems capable of harvesting biomechanical motions and other environmental stimuli including noise, wind, water, etc. Wearable electronics have captivated futuristic plots of research, and it is anticipated to achieve scalability and industrialization. Wearable sensors alone cannot fulfill the facile durable monitoring, wearable displays and LEDs can drive the futuristic view of integrating self-powered wearable sensors as point-of-care diagnostics to trace finer health details. Recently, following the green chemistry aspects, one pot synthetic conjugated block copolymer based touch a responsive LED working intelligently in achieving the wearable electronics [247]. Novel biomass-based smart orthopedics have been working intelligently with the concept of low melting polyester materials [248]. Antibacterial, breathable, and the wound healing nature of the e-skin devices can possibly grant the foundation for better wearable and disposable/degradable electronics. Smart, wearable technology has also been significant in monitoring human health such as body temperature. In compliance, Kuo et al. designed colorimetric sensors [1], and stretchable thermochromic heaters capable of producing instant visible color transformation in response to temperature [249]. Very recently, underwater self-healable electronics and its demonstration with perovskite optoelectronic device fabrication has evidenced the advancement of next generation wearable electronics [250].

Although several strategies worked satisfactorily in achieving the better figure of merit with strain and pressure sensory devices, there are still other plausible routes to explore and accomplish in near future. We have herein presented various fields of interest, demonstrating the reliability of thin film and porous and nanofibrous sensory systems. Biocompatible, eco-friendly elastomers and conducting networks that can sustain higher strains and fatigue can empower the field of strain and pressure sensors. Recently, wearable devices with good elasticity, recovery ratio, toughness, stability, hysteresis free, self-healable, adhesiveness, and breathability factors have been achieved with better balances. It is of greater importance to balance the mechanical, electrical, and ambient condition’s stability of the wearable devices to reduce the probability of mechanical and electrical failure. Fabrication methodology and simple architectures can drastically reduce the fabrication cost and time factors. By integrating greener composites, biodegradable and self-healable characters can significantly contribute to the development of next-generation sensory devices. Conducting polymer composites have a promising potential for use as stretchable and wearable sensors, field effect transistors, memory devices, LEDs, and human interactive devices to strengthen and develop the internet of things’ technological era.

## Figures and Tables

**Figure 1 polymers-13-04281-f001:**
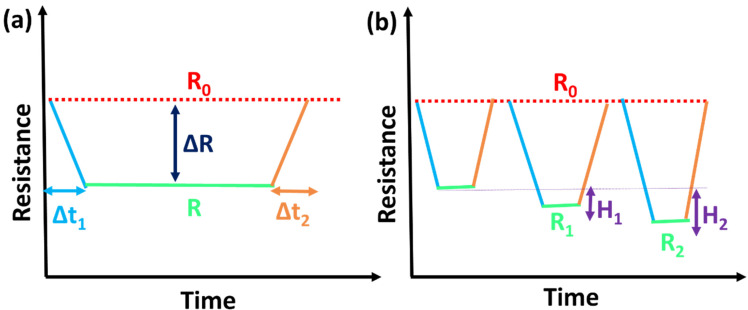
Figure of merits obtained from strain and pressure sensor response graphs, (**a**) response and recovery time, and (**b**) hysteresis among sensory response plots.

**Figure 2 polymers-13-04281-f002:**
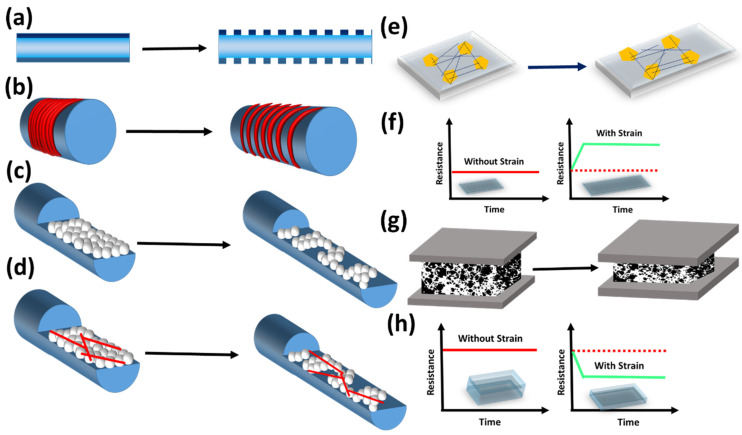
Mechanistic aspects and sensory response curve of (**a**–**f**) tensile strain sensor and (**g**,**h**) compressive pressure sensor.

**Figure 3 polymers-13-04281-f003:**
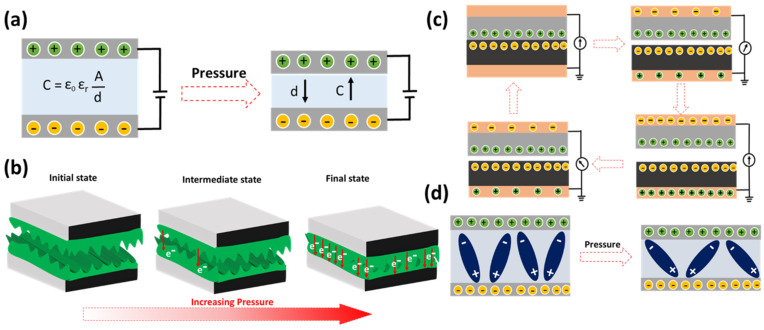
Working principle of (**a**) capacitive, (**b**) resistive, (**c**) triboelectric, and (**d**) Piezoelectric based pressure sensors.

**Figure 4 polymers-13-04281-f004:**
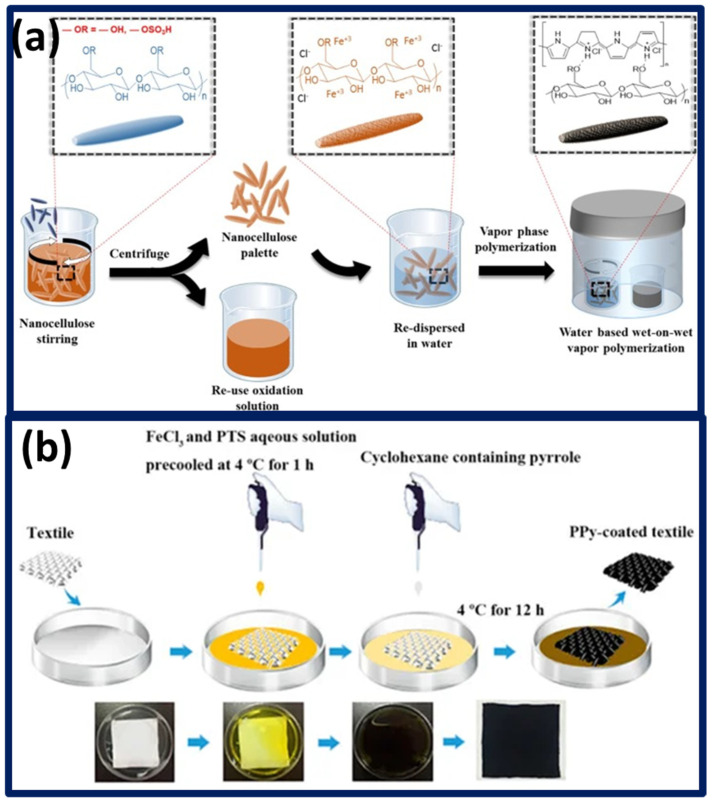
(**a**) Schematic representation of nano cellulose forming conductive structures employing vapor phase polymerization process. Reprinted from work in [85] with permission from Elsevier, 2020. (**b**) Solution-based oxidative polymerization forming the conductive textiles. Adapted from work in [87] with permission from MDPI, 2019.

**Figure 5 polymers-13-04281-f005:**
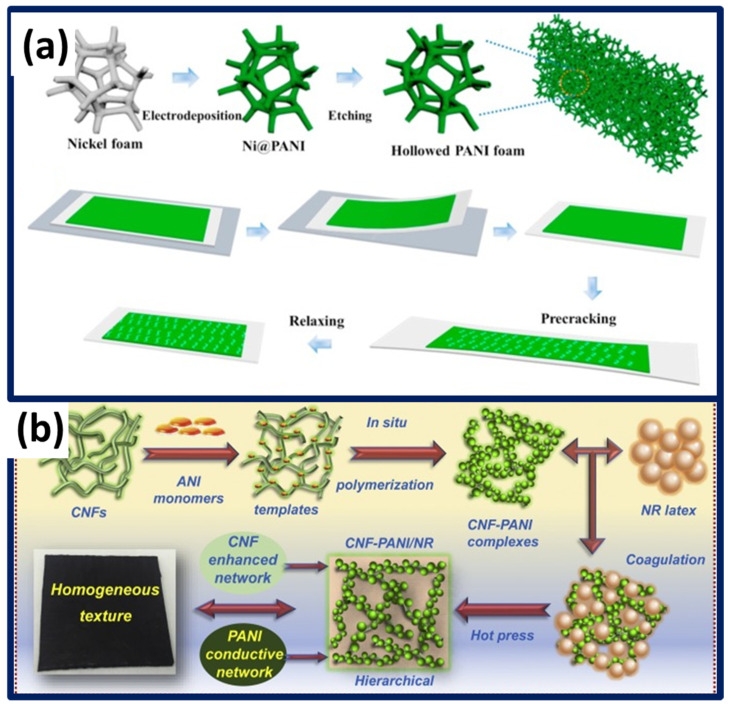
(**a**) Schematic representation to form hollow foam-like polyaniline structures with pre-cracked morphology. Reprinted from work in [124] with permission from Elsevier, 2019. (**b**) Cellulose nanofiber functionalization improving the dispersion collectively resulting in the homogenous polyaniline layer formation. Adapted from work in [131] with permission from Elsevier, 2019.

**Figure 6 polymers-13-04281-f006:**
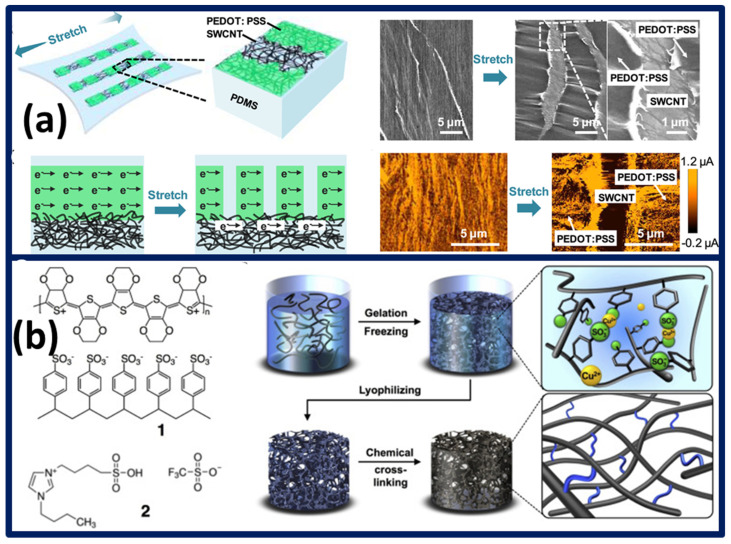
(**a**) Hierarchical conductive bridging structures supported with single-walled carbon nanotube conductive polymer composites. Reprinted from work in [189] with permission from American Chemical Society, 2020. (**b**) Freeze-drying method adopted conductive aerogel stability improvements made with chemical crosslinking. Adapted from work in [195] with permission from Elsevier, 2019.

**Figure 7 polymers-13-04281-f007:**
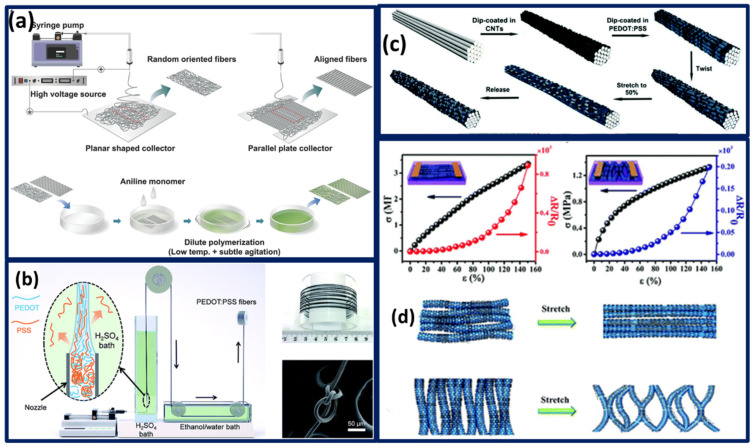
(**a**) Electrospinning random and aligned fibrous sensor fabrication process. Reprinted from work in [219] with permission from Wiley, 2018. (**b**) Wet-spinning method for the rapid fabrication of the conductive fibers via coagulation bath solution modifications. Adapted from work in [221] with permission from Royal Society of Chemistry, 2019. (**c**) Twisted polyurethane multifilament fibrous structures exhibiting micro-crack morphology. Reprinted from work in [236] with permission from Royal Society of Chemistry, 2018. (**d**) Electrospun aligned fibrous resistive response factors and its directional behavior with respective strains. Reprinted from work in [237] with permission from Royal Society of Chemistry, 2018.

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
