# Peer review of "Recent Progress in Conducting Polymer Composite/Nanofiber-Based Strain and Pressure Sensors"

_polymers, 2021, doi:10.3390/polym13244281_

Round 1
Reviewer 1 Report
The paper is far too long (even for a review paper) and resembles a book or a thesis chapter. The use of English is flawed throughout which makes the manuscript very difficult to read. There are too many instances of poor use of English to list them all in here.
I would strongly suggest that the authors restructure and reformat the paper, substantially reduce its volume and eliminate references and text that are not of primary importance to the title of the work. Hence, the entire manuscript needs to be re-written and all linguistic errors removed.
A particular concern is the equation given on line 202: C = εrA/4πkd, please note that k in this equation is the electrostatic constant, also known as Coulomb constant. k is not the dielectric constant! k=1/4πε0, where ε0=8.854×10−12 F/m is the vacuum permittivity. The authors can re-write the above equation as: C = ε0εrA/d.
Author Response
We appreciate the comments from the Academic editor and reviewers. We sincerely thank the editor and reviewers, who gave interesting comments to enhance the manuscript quality. The following is our point-to-point response to the comments.
REVIEWER 1
1. The paper is far too long (even for a review paper) and resembles a book or a thesis chapter. The use of English is flawed throughout which makes the manuscript very difficult to read. There are too many instances of poor use of English to list them all in here.
Answer: Thanks for the reviewer comments. We agree the review is too long and we eliminate few portions to make it concise. (The deleted portions are visible with track changes section) We utilized Wallace English editing service for English language and format corrections. Along with it, sequential refinements made on the manuscript to ensure the manuscript readability and language reachability. We checked our contents and polished it by having a sequence of well-defined English corrections without breaking the scientific flow. As of now, it reached its quality and there is no doubt for clear-cut readers understanding.
2. I would strongly suggest that the authors restructure and reformat the paper, substantially reduce its volume and eliminate references and text that are not of primary importance to the title of the work. Hence, the entire manuscript needs to be re-written and all linguistic errors removed.
Answer: Thanks for the reviewer comments. We collectively work on the manuscript to restructure and reformat accordingly in aim of reaching the reviewer’s expectation. We excluded several parts of the manuscript along with the reference to create better connection with reading audience. Grammatical errors and spell checks errors were eliminated, and the revised manuscript at present stage will definitely serve better in bringing light to audience without any complications.
3. A particular concern is the equation given on line 202: C = εrA/4πkd, please note that k in this equation is the electrostatic constant, also known as Coulomb constant. k is not the dielectric constant! k=1/4πε0, where ε0=8.854×10−12F/m is the vacuum permittivity. The authors can re-write the above equation as: C = ε0εrA/d.
Answer: Thanks for the reviewer valuable comments. We apologize for the equation. As per the reviewer’s suggestion, we revised the equation in the text as well as in Figure 1. Capacitive pressure sensors convert mechanical pressure signals to electrical capacitive signals. The working mode is presented in Figure 3a. The capacitance is expressed by the equation C = ε0εrA/d, where εr is the permittivity of vacuum (8.854×10−12 F/m), εr is the relative permittivity, A is the active overlap area, and d is the distance between the adjacent electrodes. (Page no.10 with no markup options in MS Word review section)
NOTE: Please kindly check the uploaded files with replying to the reviewer's comment in details (Figure). Thank you!

Reviewer 2 Report
This review is very interesting. Author comprehensively review a wide filed such the one represented by the conductive polymers. Authors presented the main sensors oriented applications. Nevertheless, this review has also two main relevant issues.
The first is the language used. It very hard to read. Due to the great efforts need to read a review, a clear languages is mandatory. Authors must deeply check sentence by sentence and improved the readability. I also suggest the use of native speaker consultancy.
The second issue is properly about the content. Authors should add a subsection prior the conclusions for the comparison of conductive polymers based materials with other available solutions. As an example, which are the advantages related to the use of conductive polymers instead of composites materials?
Anyhow, this review is promising but require a deep work to reach the quality for publication.
Author Response
We appreciate the comments from the Academic editor and reviewers. We sincerely thank the editor and reviewers, who gave interesting comments to enhance the manuscript quality. The following is our point-to-point response to the comments.
REVIEWER 2
1. This review is very interesting. Author comprehensively review a wide filed such the one represented by the conductive polymers. Authors presented the main sensors oriented applications.
Answer: Thanks for the constructive comments and we polished the manuscript even better after hearing your informative comments.
2. Nevertheless, this review has also two main relevant issues. The first is the language used. It very hard to read. Due to the great efforts need to read a review, a clear languages is mandatory. Authors must deeply check sentence by sentence and improved the readability. I also suggest the use of native speaker consultancy.
Answer: Thanks for the reviewer’s comments. We utilized Wallace English editing service for English language and format corrections. Along with it, sequential refinements made on the manuscript to ensure the manuscript readability and language reachability. We personally acknowledge the native speakers who worked collaboratively and tirelessly to promote the manuscript quality. As per the journal suggestion, we enabled the track changes options to highlight the English corrections.
3. The second issue is properly about the content. Authors should add a subsection prior the conclusions for the comparison of conductive polymers based materials with other available solutions. As an example, which are the advantages related to the use of conductive polymers instead of composites materials?
Answer: Thanks for the reviewer’s comments. According to the reviewer’s suggestion, we include a new brief subsection to highlight the advantages of conductive polymer composites over other alternatives. (Page no.54 with no markup options in MS Word review section)
- Significance of Conducting Polymer Composites Vs other alternatives
The conducting polymers and its composites offers broad spectrum of applicability with lowered fabrication complications and cost factors. Although, several breakthroughs achieved with carbon based nanostructures ranging from carbon black, graphene, carbon nanotube, carbon dots, still there is a huge space to accomplish. Still, the emergence of carbon nanostructures remains limited due to its poor dispersion, processing toxic conditions, and production cost and time factors. Variety of metallic nanostructures, alloys and metallic oxides have been taking up the challenging tasks of reigning the wearable electronics. Due to the cytotoxicity and heavy metal leakage issues, liquid metal and various alloys have not been successful in forming the human and eco-friendly wearable sensors and electronics. Conducting polymers stays unique in establishing the relationship between the device performance and fabrication due to its chemical stability, facile fabrication, low cost, biocompatible, and tunable morphology. Additionally, conducting polymer composite in recent years growing rapidly and anticipated to be the stardom in optoelectronic industry. Conducting composites serves promisingly in attaining the adhesion and durable nature on forming the conductive networks over the diverse substrates. In particular, this review highlighted the evolution and importance of conducting polymer and its composites in strain and pressure sensing applications. Conducting polymers and its composites can possibly extend its arms towards energy production, energy storage, internet of things, artificial intelligence, soft robots, artificial prosthetics, in the near future.

Round 2
Reviewer 1 Report
The revised manuscript does not address all concerns raised in my previous review.
Author Response
We appreciate the comments from the Academic editor and reviewers. We sincerely thank the editor and reviewers, who gave interesting comments to enhance the manuscript quality. The following is our point-to-point response to the comments.
NOTE: Please kindly find the attached files with all correction and detail tracking revision.
REVIEWER 1
- The paper is far too long (even for a review paper) and resembles a book or a thesis chapter. The use of English is flawed throughout which makes the manuscript very difficult to read. There are too many instances of poor use of English to list them all in here.
Answer: Thanks for the reviewer comments. We agree the review is too long and we eliminate few portions to make it concise. (The deleted portions are visible with track changes section and we added them as follows for your reference, ) We utilized Wallace English editing service for English language and format corrections. Along with it, sequential refinements made on the manuscript to ensure the manuscript readability and language reachability. We checked our contents and polished it by having a sequence of well-defined English corrections without breaking the scientific flow. As of now, it reached its quality and there is no doubt for clear-cut readers understanding.
- I would strongly suggest that the authors restructure and reformat the paper, substantially reduce its volume and eliminate references and text that are not of primary importance to the title of the work. Hence, the entire manuscript needs to be re-written and all linguistic errors removed.
Answer: Thanks for the reviewer comments. We collectively work on the manuscript to restructure and reformat accordingly in aim of reaching the reviewer’s expectation (kindly check the green highlighted text). We excluded several parts of the manuscript along with the reference to create better connection with reading audience (check the following text and pages). Grammatical errors and spell checks errors were eliminated, and the revised manuscript at present stage will definitely serve better in bringing light to audience without any complications.
- A particular concern is the equation given on line 202: C = εrA/4πkd, please note that k in this equation is the electrostatic constant, also known as Coulomb constant. k is not the dielectric constant! k=1/4πε0, where ε0=8.854×10−12F/m is the vacuum permittivity. The authors can re-write the above equation as: C = ε0εrA/d.
Answer: Thanks for the reviewer valuable comments. We apologize for the equation. As per the reviewer’s suggestion, we revised the equation in the text as well as in Figure 1.
Capacitive pressure sensors convert mechanical pressure signals to electrical capacitive signals. The working mode is presented in Figure 3a. The capacitance is expressed by the equation C = ε0εrA/d, where εr is the permittivity of vacuum (8.854×10−12 F/m), εr is the relative permittivity, A is the active overlap area, and d is the distance between the adjacent electrodes. (Page no.10 with no markup options in MS Word review section)

Reviewer 2 Report
i endorse the publication of this paper without any further modifications
Author Response
Than you so much for your agree that our revised manuscript can be accepted and published on Polymers without any further modifications. Thank you for your suggestion.
Round 3
Reviewer 1 Report
Please note that ε0=8.854×10−12F/m is the vacuum permittivity.
The use of English is still unacceptable for a publication.
Author Response
We appreciate the comments from the Academic editor and reviewers. We sincerely thank the editor and reviewers, who gave interesting comments to enhance the manuscript quality. The following is our point-to-point response to the comments.
Reviewer 1 Comments
- Comments and suggestions for Authors. Please note that ε0=8.854×10−12F/m is the vacuum permittivity.
Answer: Thanks for the comments. We made corrections as per your suggestions and it is presented as follows,
Capacitive pressure sensors convert mechanical pressure signals to electrical capacitive signals. The working mode is presented in Figure 3a. The capacitance is expressed by the equation C = ε0εrA/d, where ε0 is the vacuum permittivity (8.854×10−12 F/m), εr is the relative permittivity, A is the active overlap area, and d is the distance between the adjacent electrodes [64]. (Page no.11)
- The use of English is still unacceptable for a publication.
Answer: Thanks for the comments. First stage refinements made on typo and grammatical errors. Sequential revisions from native English speakers and Wallace English editing service providers effectively strengthened the quality of the revised manuscript. At present stage, this review article will definitely serve better in bringing light to audience without any complications.
Thank you very much for the editor and the reviewers’ time and patience to read this response letter. We really appreciate your valuable suggestions and comments. We hope this revised manuscript will meet with approval.
